# Two Facets of SDE Under an Information-Theoretic Lens: Generalization of SGD via Training Trajectories and via Terminal States

## Abstract

Stochastic differential equations (SDEs) have been shown recently to well characterize the dynamics of training machine learning models with SGD. This provides two opportunities for better understanding the generalization behaviour of SGD through its SDE approximation. Firstly, viewing SGD as full-batch gradient descent with Gaussian gradient noise allows us to obtain trajectories-based generalization bound using the information-theoretic bound from Xu & Raginsky (2017). Secondly, assuming mild conditions, we estimate the steady-state weight distribution of SDE and use information-theoretic bounds from Xu & Raginsky (2017) and Negrea et al. (2019) to establish terminal-state-based generalization bounds. Our proposed bounds have some advantages, notably the trajectories-based bound outperforms results in Wang & Mao (2022), and the terminal-state-based bound exhibits a fast decay rate comparable to stability-based bounds.

## 1 Introduction

Modern deep neural networks trained with SGD and its variants have achieved surprising successes: the overparametrized networks often contain more parameters than the size of training dataset, yet capable of generalizing well on the testing set; this contrasts the traditional wisdom in statistical learning that suggests such high-capacity models will overfit the training data and fail on the unseen data (Zhang et al., 2017). Intense recent efforts have been spent to explain this peculiar phenomenon via investigating the properties of SGD (Arpit et al., 2017; Bartlett et al., 2017; Neyshabur et al., 2017; Arora et al., 2019), and the current understanding is still far from being complete. For example, neural tangent kernel (NTK) based generalization bounds of SGD normally require the width of network to be sufficiently large (or even go to infinite) (Arora et al., 2019), and the stability based bounds of SGD have a poorly dependence on an intractable Lipschitz constant (Hardt et al., 2016).

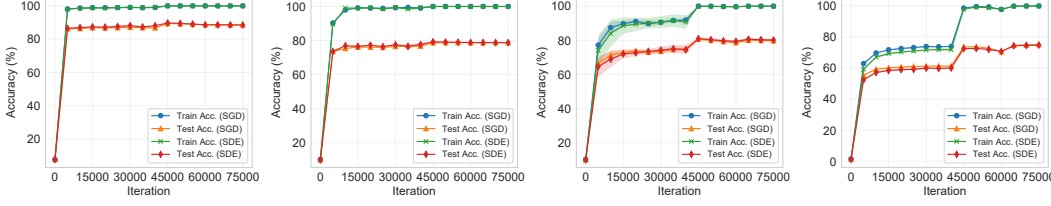

| (a) VGG on (small) SVHN | (b) VGG on CIFAR10 | (c) ResNet on CIFAR10 | (d) ResNet on CIFAR100 |

Figure 1: Performance of VGG-11 and ResNet-18 trained with SGD and SDE.

Recently, information-theoretic generalization bounds have been developed to analyze the expected generalization error of a learning algorithm. The main advantage of such bounds is that they are not only distribution-dependent, but also algorithm-dependent, making them an ideal tool for studying the generalization behaviour of models trained with a specific algorithm, such as SGD. Mutual information (MI) based bounds are first proposed by (Russo & Zou, 2016; 2019; Xu & Raginsky, 2017). They are then strengthened by additional techniques (Asadi et al., 2018; Negrea et al., 2019;

Bu et al., 2019; Steinke & Zakynthinou, 2020; Haghifam et al., 2020; Wang et al., 2021b). Particularly, Negrea et al. (2019) derive MI-based bounds by developing a PAC-Bayes-like bounding technique, which upper-bounds the generalization error in terms of the KL divergence between the posterior distribution of learned model parameter given by a learning algorithm with respect to any data-dependent prior distribution. It is remarkable that the application of these information-theoretic techniques usually requires the learning algorithm to be an iterative noisy algorithm, such as stochastic gradient Langevin dynamics (SGLD) (Raginsky et al., 2017; Pensia et al., 2018), so as to avoid the MI bounds becoming infinity, and can not be directly applied to SGD. In order to apply such techniques to SGD, Neu et al. (2021) and Wang & Mao (2022) develop generalization bounds for SGD via constructing an auxiliary iterative noisy process. However, identifying an optimal auxiliary process is difficult, and arbitrary choices may not provide meaningful insights into the generalization of SGD, see Appendix B.1 for more discussions.

Recent research has suggested that the SGD dynamics can be well approximated by using stochastic differential equations (SDEs), where the gradient signal in SGD is regarded as the full-batch gradient perturbed with an additive Gaussian noise. Specifically, Mandt et al. (2017) and Jastrzębski et al. (2017) model this gradient noise drawn from a Gaussian distribution with a fixed covariance matrix, thereby viewing SGD as performing variational inference. (Zhu et al., 2019; Wu et al., 2020; Xie et al., 2021a;b) further model the gradient noise as dependent of the current weight parameter and the training data. Moreover, Li et al. (2017; 2019) and Wu et al. (2020) prove that when the learning rate is sufficiently small, the SDE trajectories are theoretically close to those of SGD (see Lemma B.1). More recently, Li et al. (2021) has demonstrated that the SDE approximation well characterizes the optimization and generalization behavior of SGD without requiring small learning rates.

In this work, we also empirically verify the consistency between the dynamics of SGD and its associated discrete SDE (i.e. Eq. (5)). As illustrated in Figure 1, the strong agreement in their performance suggests that, despite the potential presence of non-Gaussian components in the SGD gradient noise, analyzing its SDE through a Gaussian approximation suffices for exploring SGD's generalization behavior. Furthermore, under the SDE formalism of SGD, SGD becomes an iterative noisy algorithm, on which the aforementioned information-theoretic bounding techniques can directly apply. In particular, we summarize our contributions below.

- We obtain a generalization bound (Theorem 3.1) in the form of a summation over training steps of a quantity that involves both the sensitivity of the full-batch gradient to the variation of the training set and the covariance of the gradient noise (which makes the SGD gradient deviate from the full-batch gradient). We also give a tighter bound in Theorem 3.2, where the generalization performance of SGD depends on the alignment of the population gradient covariance and the batch gradient covariance. These bounds highlight the significance of (the trace of) the gradient noise covariance in the generalization ability of SGD.

- In addition to the time-dependent trajectories-based bounds, we also provide time-independent (or asymptotic) bounds by some mild assumptions. Specifically, based on previous information-theoretic bounds, we obtain generalization bounds in terms of the KL divergence between the steady-state weight distribution of SGD with respect to a distribution-dependent prior distribution (by Lemma 2.1) or data-dependent prior distribution (by Lemma 2.2). The former gives us a bound based on the alignment between the weight covariance matrix for each individual local minimum and the weight covariance matrix for the average of local minima (Theorem 4.1). Under mild assumptions, we can estimate the steady-state weight distribution of SDE (Lemma 4.1), leading to a variant of Theorem 4.1 (Corollary 4.1) and a norm-based bound (Corollary 4.2). Additionally, we obtain a stability-based like bound by Lemma 2.2 (Theorem 4.2), with the notable omission of the Lipschitz constant in other stability-based bounds. Since stability-based bounds often achieve fast decay rates, e.g., $\mathcal{O}(1/n)$, Theorem 4.2 provides theoretical advantages compared with other information-theoretic bounds, as it can attain the same rate of decay as the stability-based bound. Comparing to the first family of bounds (i.e., trajectories-based bounds), the second family of bounds directly bound the generalization error via the terminal state, which avoids summing over training steps; these bounds can be tighter when the steady-state estimates are accurate. On the other hand, not relying on the steady-state estimates and the approximating assumptions they base upon is arguably an advantage of the first family.

- We empirically analyze key components within the derived bounds for both algorithms. Our empirical findings reveal that these components for SGD and SDE align remarkably well, further

validating the effectiveness of our bounds for assessing the generalization of SGD. Moreover, we provide numerical validation of the presented bounds and demonstrate that our trajectories-based bound is tighter than the result in Wang & Mao (2022). Additionally, compared with norm-based bounds, we show that the terminal-state-based bound that integrates the geometric properties of local minima can better characterize generalization.

**Other Related Literature** Recently, (Simsekli et al., 2019; Nguyen et al., 2019; Simsekli et al., 2020; Meng et al., 2020; Gurbuzbalaban et al., 2021) challenge the traditional assumption that gradient noise is a Gaussian and argue that the noise is heavy-tailed (e.g., Lévy noise). In contrast, Xie et al. (2021a) and Li et al. (2021) claim that non-Gaussian noise is not essential to SGD performance, and SDE with Gaussian gradient noise can well characterize the behavior of SGD. They also argue that the empirical evidence shown in Simsekli et al. (2019) relies on a hidden strong assumption that gradient noise is isotropic and each dimension has the same distribution. Other works on SGD and SDE, see (Hoffer et al., 2017; Xing et al., 2018; Panigrahi et al., 2019; Wu et al., 2020; Zhu et al., 2019; Li et al., 2020; Ziyin et al., 2022).

## 2 PRELIMINARY

**Expected Generalization Error** Let $\mathcal{Z}$ be the instance space and let $\mu$ be an unknown distribution on $\mathcal{Z}$, specifying random variable $Z$. We let $\mathcal{W} \subseteq \mathbb{R}^d$ be the space of hypotheses. In the information-theoretic analysis framework, there is a training sample $S = \{Z_1, Z_2, \ldots, Z_n\}_{i=1}^n$ drawn i.i.d. from $\mu$ and a stochastic learning algorithm $\mathcal{A}$ takes the training sample $S$ as its input and outputs a hypothesis $W \in \mathcal{W}$ according to some conditional distribution $Q_{W|S}$. Given a loss function $\ell : \mathcal{W} \times \mathcal{Z} \to \mathbb{R}^+$, where $\ell(w, z)$ measures the "unfitness" or "error" of any $z \in \mathcal{Z}$ with respect to a hypothesis $w \in \mathcal{W}$. The goal of learning is to find a hypothesis $w$ that minimizes the population risk, and for any $w \in \mathcal{W}$, the population risk is defined as $L_\mu(w) \triangleq \mathbb{E}_{Z \sim \mu}[\ell(w, Z)]$. In practice, since $\mu$ is only partially accessible via the sample $S$, we instead turn to use the empirical risk, defined as $L_S(w) \triangleq \frac{1}{n} \sum_{i=1}^n \ell(w, Z_i)$. Then, the expected generalization error of $\mathcal{A}$ is defined as $\mathcal{E}_\mu(\mathcal{A}) \triangleq \mathbb{E}_{W,S}[L_\mu(W) - L_S(W)]$, where the expectation is taken over $(S, W) \sim \mu^n \otimes Q_{W|S}$.

Throughout this paper, we assume that $\ell$ is differentiable almost everywhere with respect to $w$. In some cases we will assume that $\ell(w, Z)$ is $R$-subgaussian for any $w \in \mathcal{W}$. Note that a bounded loss is guaranteed to be subgaussian. We will denote $\ell(w, Z_i)$ by $\ell_i$ when there is no ambiguity.

**SGD and SDE** At each time step $t$, given the current state $W_{t-1} = w_{t-1}$, let $B_t$ be a random subset that is drawn uniformly from $\{1, 2, \ldots, n\}$ and $|B_t| = b$ is the batch size. Let $\widetilde{G}_t \triangleq \frac{1}{b} \sum_{i \in B_t} \nabla \ell_i$ be the mini-batch gradient. The SGD updating rule with learning rate $\eta$ is then

$$W_t = w_{t-1} - \eta \widetilde{G}_t. \tag{1}$$

The full batch gradient is $G_t \triangleq \frac{1}{n} \sum_{i=1}^n \nabla \ell_i$. It follows that

$$W_t = w_{t-1} - \eta G_t + \eta V_t, \tag{2}$$

where $V_t \triangleq G_t - \widetilde{G}_t$ is the mini-batch *gradient noise*. Since $\mathbb{E}_{B_t}[V_t] = 0$, $\widetilde{G}_t$ is an unbiased estimator of the full batch gradient $G_t$. Moreover, the single-draw (i.e. $b = 1$) SGD gradient noise covariance (GNC) and the mini-batch GNC are $\Sigma_t = \frac{1}{n} \sum_{i=1}^n \nabla \ell_i \nabla \ell_i^T - G_t G_t^T$ and $C_t = \frac{n-b}{b(n-1)} \Sigma_t$, respectively. If $n \gg b$, then $C_t = 1/b\Sigma_t$. Notice that $\Sigma_t$ (or $C_t$) is state-dependent, i.e. it depends on $w_{t-1}$. If $t$ is not specified, we use $\Sigma_w$ (or $C_w$) to represent its dependence on $w$. In addition, the population GNC at time $t$ is

$$\Sigma_t^\mu \triangleq \mathbb{E}_Z \left[ \nabla \ell(w_{t-1}, Z) \nabla \ell(w_{t-1}, Z)^T \right] - \mathbb{E}_Z \left[ \nabla \ell(w_{t-1}, Z) \right] \mathbb{E}_Z \left[ \nabla \ell(w_{t-1}, Z)^T \right]. \tag{3}$$

We assume that the initial parameter $W_0$ is independent of all other random variables, and SGD stops after $T$ updates, outputting $W_T$ as the learned parameter.

We now approximate $V_t$ up to its second moment, e.g., $V_t \sim \mathcal{N}(0, C_t)$, then we have the following continuous-time evolution, i.e. Itô SDE:

$$d\omega = -\nabla L_S(\omega)dt + [\eta C_\omega]^{\frac{1}{2}} d\theta_t, \tag{4}$$

where $C_\omega$ is the GNC at $\omega$ and $\theta_t$ is a Wiener process. Furthermore, the *Euler-Maruyama* discretization, as the simplest approximation scheme to Itô SDE in Eq. (4), is

$$W_t = w_{t-1} - \eta G_t + \eta C_t^{1/2} N_t, \tag{5}$$

where $N_t \sim \mathcal{N}(0, \mathrm{I}_d)$ is the standard Gaussian random variable.

**Validation of SDE** It is important to understand how accurate of SDE in Eq. (4) for approximating the SGD process in Eq. (1). Previous research, such as (Li et al., 2017; 2019), has provided theoretical evidence supporting the idea that SDE can approximate SGD in a "weak sense". That is, the SDE processes closely mimic the original SGD processes, not on an individual sample path basis, but rather in terms of their distributions (see Lemma B.1 for a formal result).

Additionally, concerning the validation of the discretization of SDE in Eq. (5), Wu et al. (2020, Theorem 2) has proved that Eq. (5) is *an order* 1 *strong approximation* to SDE in Eq. (4). Moreover, we direct interested readers to the comprehensive investigations carried out by (Wu et al., 2020; Li et al., 2021), where the authors empirically verify that SGD and Eq. (5) can achieve the similar testing performance, suggesting that non-Gaussian noise is not essential to SGD performance. In other words, studying Eq. (5) is arguably sufficient to understand generalization properties of SGD. In Figure 1, we also empirically verify the approximation of Eq. (5), and show that it can effectively capture the behavior of SGD.

**Two Information-Theoretic Bounds** The original version of mutual information based bound is a sample-based MI bound whose main component is the mutual information between the output $W$ and the entire input sample $S$. This result is given as follows:

**Lemma 2.1** (Xu & Raginsky (2017, Theorem 1.))**.** *Assume the loss $\ell(w, Z)$ is R-subGaussian for any $w \in \mathcal{W}$, then $|\mathcal{E}_\mu(\mathcal{A})| \leq \sqrt{\frac{2R^2}{n} I(W; S)}$.*

This bound is further improved by a data-dependent prior based bound. Following the setup in Negrea et al. (2019), let $J$ be a random subset uniformly drawn from $\{1, \ldots, n\}$ and $|J| = m > b$. Let $S_J = \{Z_i\}_{i \in J}$. Typically, we choose $m = n - 1$, then the following result is known.

**Lemma 2.2** (Negrea et al. (2019, Theorem 2.5))**.** *Assume the loss $\ell(w, Z)$ is bounded in $[0, M]$, then for any $P_{W|S_J}$, $\mathcal{E}_\mu(\mathcal{A}) \leq \frac{M}{\sqrt{2}} \mathbb{E}_{S,J} \sqrt{D_{\mathrm{KL}}(Q_{W|S} || P_{W|S_J})}$.*

Note that $J$ is drawn before the training starts and is independent of $\{W_t\}_{t=0}^T$. We use the subset $S_J$ to conduct a parallel SGD training process based to obtain a data-dependent prior ($P_{W|S_J}$). When $m = n - 1$, we call this prior process the leave-one-out (LOO) prior.

## 3 GENERALIZATION BOUNDS VIA FULL TRAJECTORIES

We now discuss the generalization of SGD under the approximation of Eq. (5). In particular, we let $\widehat{G}_t = -G_t + C_t^{1/2} N_t$. We first have the following lemma.

**Lemma 3.1.** $I(\widehat{G}_t; S|W_{t-1}) = \mathbb{E}_{W_{t-1}} \left[ \inf_{P_{\widehat{G}_t|W_{t-1}}} \mathbb{E}_S^{W_{t-1}} \left[ D_{\mathrm{KL}}(Q_{\widehat{G}_t|S,W_{t-1}} || P_{\widehat{G}_t|W_{t-1}}) \right] \right]$, *where the infimum is achieved when the prior distribution $P_{\widehat{G}_t|w_{t-1}} = Q_{\widehat{G}_t|w_{t-1}}$ for any t.*

Lemma 3.1 suggests that every choice of $P_{\widehat{G}_t|W_{t-1}}$ gives rise to an upper bound of the MI of interest via $I(\widehat{G}_t; S|W_{t-1}) \leq \mathbb{E}_{W_{t-1}} \left[ \mathbb{E}_S^{W_{t-1}} \left[ D_{\mathrm{KL}}(Q_{\widehat{G}_t|S,W_{t-1}} || P_{\widehat{G}_t|W_{t-1}}) \right] \right]$. The closer is $P_{\widehat{G}_t|W_{t-1}}$ to $Q_{\widehat{G}_t|W_{t-1}}$, the tighter is the bound. As the simplest choice, we will first choose an isotropic Gaussian prior, $P_{\widehat{G}_t|w_{t-1}} = \mathcal{N}(\tilde{g}_t, \sigma_t^2 I_d)$ (where both $\tilde{g}_t$ and $\sigma_t$ are only allowed to depend on $W_{t-1}$), and optimize the KL divergence in Lemma 3.1 over $\sigma_t$ for a fixed $\tilde{g}_t$. The following result is obtained.

**Theorem 3.1.** *Under the conditions of Lemma 2.1 and assume $C_t$ is a positive-definite matrix. For any $t \in [T]$, let $\tilde{g}_t$ be any constant vector for a given $w_{t-1}$, then*

$$\mathcal{E}_\mu(\mathcal{A}) \leq \sqrt{\frac{R^2}{n} \sum_{t=1}^T \mathbb{E}_{W_{t-1}} \left[ d \log \frac{h_1(W_{t-1})}{d} - h_2(W_{t-1}) \right]}, \tag{6}$$

*where $h_1(w) = \mathbb{E}_S^w \left[ ||G_t - \tilde{g}_t||^2 + tr\{C_t\} \right]$ and $h_2(w) = \mathbb{E}_S^w [tr\{\log C_t\}]$.*

*Furthermore, if $\tilde{g}_t = \mathbb{E}_Z [\nabla \ell(w_{t-1}, Z)]$, then $h_1(w) = \frac{1}{b} tr\{\Sigma_t^\mu\}$.*

Notice that $\tilde{g}_t$ is any reference "gradient" independent of $S$, then the first term in $h_1(W_{t-1})$, $||G_t - \tilde{g}_t||^2$, characterizes the sensitivity of the full-batch gradient to some variation of the training

set $S$, while the second term in $h_1(W_{t-1})$, i.e. $tr\{C_t\}$, reflects the gradient noise magnitude induced by the mini-batch based training. For example, if $\tilde{g}_t = \mathbb{E}_Z[\nabla \ell(w_{t-1}, Z)]$, then $\mathbb{E}_S^{w_{t-1}}\left[\|G_t - \tilde{g}_t\|^2\right]$ is the variance of the gradient sample mean, and such $\tilde{g}_t$ will eventually convert $h_1(W_{t-1})$ to the population GNC, namely $h_1(W_{t-1}) = \frac{1}{b}tr\{\Sigma_t^\mu\}$.

Moreover, if we simply let $\tilde{g}_t = 0$, then Theorem 3.1 indicates that one can control the generalization performance via controlling the gradient norm along the entire training trajectories, e.g., if we further let $b = 1$, then $h_1(W_{t-1}) = \frac{1}{n}\sum_{i=1}^n \|\nabla \ell_i\|^2$. This is consistent with the existing practice, for example, applying gradient clipping (Wang & Mao, 2022; Geiping et al., 2022) and gradient penalty (Jastrzebski et al., 2021; Barrett & Dherin, 2021; Smith et al., 2021; Geiping et al., 2022) as regularization techniques to improve generalization.

As a by-product, we recover previous information-theoretic bounds for the Gradient Langevin dynamics (GLD) with noise distribution $\mathcal{N}(0, \eta^2 \mathrm{I}_d)$ below.

**Corollary 3.1.** *If $C_t = \mathrm{I}_d$, then $\mathcal{E}_\mu(\mathcal{A}) \leq \sqrt{\frac{R^2 d}{n}\sum_{t=1}^T \mathbb{E}_{W_{t-1}}\log\left(\mathbb{E}_S^{W_{t-1}}\|G_t - \tilde{g}_t\|^2/d + 1\right)}$.*

Note that the bound in Corollary 3.1 is tighter than the bound in Neu et al. (2021, Proposition 3.) by $\log(x+1) \leq x$ and it is also stronger than the bound in Pensia et al. (2018) due to the fact that we use a state dependent quantity $\mathbb{E}_S^{w_{t-1}}\left[\|G_t - \tilde{g}_t\|^2\right]$ rather than some global Lipschitz constant.

While choosing the isotropic Gaussian prior is common in the GLD or SGLD setting, given that we already know $C_t$ is an anisotropic covariance, one can select an anisotropic prior to better incorporate the geometric structure in the prior distribution. A natural choice of the covariance is a scaled population GNC, namely $\tilde{c}_t \Sigma_t^\mu$, where $\tilde{c}_t$ is some positive state dependent scaling factor. Let $\tilde{g}_t = \mathbb{E}_Z[\nabla \ell(w_{t-1}, Z)]$ be the state dependent mean. By optimizing over $c_t$, we have the bound below.

**Theorem 3.2.** *Under the conditions of Lemma 2.1 and assume $C_t$ and $\Sigma_t^\mu$ are positive-definite matrices, then $\mathcal{E}_\mu(\mathcal{A}) \leq \sqrt{\frac{R^2}{n}\sum_{t=1}^T \mathbb{E}_{W_{t-1},S}\left[tr\left\{\log\frac{\Sigma_t^\mu C_t^{-1}}{b}\right\}\right]}$.*

**Remark 3.1.** *If we let the diagonal element of $\Sigma_t^\mu$ in dimension $k$ be $\alpha_t(k)$ and let the corresponding diagonal element of $\Sigma_t$ be $\beta_t(k)$, and assume $n \gg b$ (so $\Sigma_t = bC_t$), then $tr\{\log(\Sigma_t^\mu C_t^{-1}/b)\} = \sum_{k=1}^d \log\frac{\alpha_t(k)}{\beta_t(k)}$. Thus, Theorem 3.2 implies that a favorable alignment between the diagonal values of $\Sigma_t$ and $\Sigma_t^\mu$ will positively impact generalization performance. In other words, the perfect alignment of these two matrices indicates that SGD is insensitive to the randomness of $S$. Recall the key quantity in Lemma 2.1, $I(W; S)$, which also measures the dependence of $W$ with the randomness of $S$, the term $\Sigma_t^\mu \Sigma_t^{-1}$ conveys a similar intuition in this context.*

Compared with Theorem 3.1 under the same choice of $\tilde{g}_t$, we notice that the main difference is that the term $tr\{\log(\Sigma_t^\mu/b)\}$, instead of $d\log(tr\{\Sigma_t^\mu\}/bd)$, appears in the bound of Theorem 3.2. The following lemma demonstrates that Theorem 3.2 is tighter than the bound in Theorem 3.1.

**Lemma 3.2.** *For any $t$, we have $tr\left\{\log\frac{\Sigma_t^\mu}{b}\right\} \leq d\log\frac{tr\{\Sigma_t^\mu\}}{bd}$, with the equality holds when all the diagonal elements in $\Sigma_t^\mu$ have the same value, i.e. $\alpha_t(1) = \alpha_t(2) = \cdots = \alpha_t(d)$.*

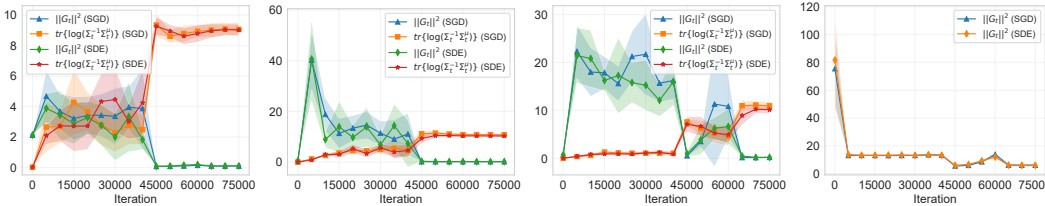

(a) VGG on (small) SVHN    (b) VGG on CIFAR10    (c) ResNet on CIFAR10    (d) ResNet on CIFAR100

Figure 2: Gradient-related quantities of SGD or its discrete SDE approximation. In (d), since per-sample gradient is ill-defined when BatchNormalization is used, we do not track $tr\left\{\log\left(\Sigma_t^{-1}\Sigma_t^\mu\right)\right\}$. The trajectories-based bounds in Theorem 3.1 and Theorem 3.2 emphasize the significance of gradient-related information along entire trajectories, including metrics such as gradient norm and

gradient covariance alignment, in comprehending the generalization dynamics of understanding the generalization of SGD. In Figure 2, we visually show that these key gradient-based measures during SDE training closely mirror the dynamics observed in SGD.

Notably, these trajectory-based information-theoretic bounds are time-dependent, indicating that these bounds may grow with the training iteration number $T$, unless the gradient norm becomes negligible at some point during training. While the stability-based bounds for GD/SGD are also time-dependent (Hardt et al., 2016; Bassily et al., 2020) (in the convex learning case), the learning rate in these bounds helps mitigate the growth of $T$. However, the learning rate does not appear in our trajectory-based information-theoretic bounds, making the dependency on $T$ even worse.

Note that Wang et al. (2021b) uses the strong data processing inequality to reduce this deficiency, but the bound still increases with $T$. To tackle this weakness, we will invoke some asymptotic SDE results on the terminal parameters of the algorithm, which will give us a crisp way to characterize the expected generalization gap without decomposing the mutual information.

## 4 GENERALIZATION BOUNDS VIA TERMINAL STATE

In this section, we directly bound the generalization error by the properties of the terminal state instead of using the full training trajectory information. Particularly, we will first use the stationary distribution of weights at the end of training as $Q_{W_T|S}$. To overcome the explicit time-dependence present in the bounds discussed in Section 3, one must introduce additional assumptions, with these assumptions being the inherent cost. For example, an important approximation used in this section is the quadratic approximation of the loss. Specifically, let $w_s^*$ be a local minimum for a given training sample $S = s$, when $w$ is close to $w_s^*$, we can use a second-order Taylor expansion to approximate the value of the loss at $w$,

$$L_s(w) = L_s(w_s^*) + \frac{1}{2}(w - w_s^*)^{\mathbf{T}} H_{w_s^*}(w - w_s^*). \tag{7}$$

where $H_{w_s^*}$ is the Hessian matrix of $s$ at $w^*$. Note that in this case, when $w_t \to w_s^*$, we have $G_t = \nabla L_s(w_t) = H_{w_s^*}(w_t - w_s^*)$. Our remaining analysis assumes the validity of Eq. (7).

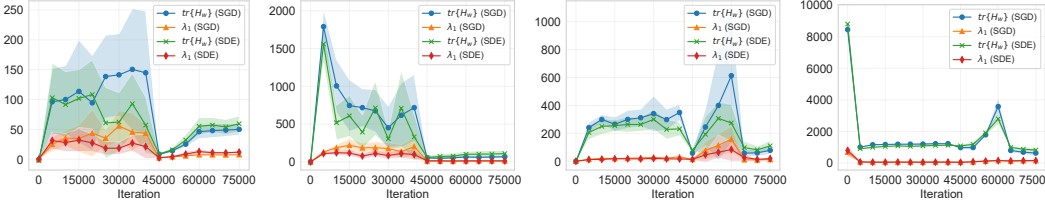

(a) VGG on (small) SVHN    (b) VGG on CIFAR10    (c) ResNet on CIFAR10    (d) ResNet on CIFAR100

Figure 3: Hessian-related quantities of SGD or its discrete SDE approximation.

In view of Eq. (7), a classical result of Mandt et al. (2017) shows that the posterior $Q_{W|s}$ around $w_s^*$ is a Gaussian distribution $\mathcal{N}(w_s^*, \Lambda_{w_s^*})$, where $\Lambda_{w^*} \triangleq \mathbb{E}\left[(W - w^*)(W - w^*)^T\right]$ is the covariance of the stationary distribution (see Appendix B.3 for an elaboration).

In the context of nonconvex learning, e.g., deep learning, we usually have more than one local minimum $w_s^*$ for a give $S = s$. Hence, it is necessary to take the local minimum itself as a random variable, namely $W_s^* \sim Q_{W_s^*|s}$. In this case, we have $Q_{W|s,w_s^*} = \mathcal{N}(w_s^*, \Lambda_{w_s^*})$ and the posterior distribution $Q_{W|s} = \mathbb{E}_{W_s^*}^s\left[\mathcal{N}(W_s^*, \Lambda_{W_s^*})\right]$ should be a mixture of Gaussian distributions. Moreover, recall that $I(W_T; S) = \inf_{P_{W_T}} \mathbb{E}_S D_{\mathrm{KL}}(Q_{W_T|S}||P_{W_T})$ where $P_{W_T} = Q_{W_T}$ achieves the infimum. Here the oracle prior $Q_{W_T} = \mathbb{E}_{S,W_S^*}\left[\mathcal{N}(W_S^*, \Lambda_{W_S^*})\right]$ is also a mixture of Gaussian distributions. From a technique side, given that the KL divergence between two mixture of Gaussian distributions does not have an analytic closed form, we turn to analyze its upper bound, namely $\inf_{P_{W_T}} \mathbb{E}_{S,W_S^*} D_{\mathrm{KL}}(Q_{W_T|S,W_S^*}||P_{W_T})$. When each $s$ only has one local minimum, $I(W; S)$ will be equal to this upper bound.

We are ready to give the terminal-state-based bounds.

**Theorem 4.1.** *Let $w_\mu^* = \mathbb{E}[W_S^*]$ be the expected ERM solution and let $\Lambda_{w_\mu^*} = \mathbb{E}\left[\left(W_T - w_\mu^*\right)\left(W_T - w_\mu^*\right)^{\mathbf{T}}\right]$ be its corresponding stationary covariance, then $\mathcal{E}_\mu(\mathcal{A}) \leq \frac{R}{\sqrt{2n}}\sqrt{\mathbb{E}_{S,W_S^*}\left[tr\left\{\log\left(\Lambda_{W_S^*}^{-1}\Lambda_{w_\mu^*}\right)\right\}\right]}.$*

This result bears resemblance to Theorem 3.2 since both involve the alignment between a population covariance matrix and a sample (or batch) covariance matrix.

Note that $\Lambda_{w_\mu^*} = \mathbb{E}\left[\left(W_S^* - w_\mu^*\right)\left(W_S^* - w_\mu^*\right)^{\mathbf{T}}\right] + \mathbb{E}\left[\Lambda_{W_S^*}\right]$. By Jensen's inequality, we can bring the expectation over $W_s^*$ inside the logarithmic function. Additionally, if $\mathbb{E}_{W_s^*}\left[\Lambda_{W_s^*}^{-1}\mathbb{E}\left[\Lambda_{W_s^*}\right]\right]$ is close to the identity matrix—especially evident in scenarios where each $s$ has only one minimum, as in convex learning—we obtain the upper bound $\mathcal{O}\left(\sqrt{\mathbb{E}\left[d_M^2\left(W_S^*, w_\mu^*; \Lambda_{W_S^*}\right)\right]/n}\right)$, where $d_M(x, y; \Sigma) \triangleq \sqrt{(x-y)^T\Sigma^{-1}(x-y)}$ is the Mahalanobis distance. Intuitively, this quantity measures the sensitivity of a local minimum to the combined randomness introduced by both the algorithm and the training sample, relative to its local geometry.

In practice, one can estimate $\Lambda_{w_\mu^*}$ and $\Lambda_{w_s^*}$ by repeatedly conducting training processes and storing numerous checkpoints at the end of each training run. This is still much easier than estimating $I(W; S)$ directly. As an alternative strategy, one may leverage the analytical expression available for $\Lambda_{w_s^*}$. Mandt et al. (2017) provides such analysis and give a equation to solve for $\Lambda_{w_s^*}$. However, the result in Mandt et al. (2017) relies on the unrealistic small learning rate and the GNC in their analysis is regarded as a state-independent covariance matrix. To overcome these limitations, we give the following result under a quadratic approximation of the loss, which is refined from Liu et al. (2021, Theorem 1.) by using the state-dependent GNC.

**Lemma 4.1.** *In the long term limit, we have $\Lambda_{w^*}H_{w^*} + H_{w^*}\Lambda_{w^*} - \eta H_{w^*}\Lambda_{w^*}H_{w^*} = \eta C_T$. Moreover, consider the conditions: (i) $H_{w^*}\Lambda_{w^*} = \Lambda_{w^*}H_{w^*}$; (ii) $H_{w^*}^{-1}\Sigma_T = I_d$; (iii) $\frac{2}{\eta} \gg \lambda_1$ where $\lambda_1$ is the top-1 eigenvalue of $H_{w^*}$. Hence, given (i), we have $\Lambda_{w^*} = \left[H_{w^*}\left(\frac{2}{\eta}I_d - H_{w^*}\right)\right]^{-1}C_T$; given (i-ii), we have $\Lambda_{w^*} = (\frac{2}{\eta}I_d - H_{w^*})^{-1}$; given (i-iii), we have $\Lambda_{w^*} = \frac{\eta}{2b}I_d$.*

Notably, all the conditions in Lemma 4.1 are only discussed in the context of the terminal state of SGD training. Regarding the condition (ii), as being widely used in the literature (Jastrzębski et al., 2017; Zhu et al., 2019; Li et al., 2020; Xie et al., 2021a;b; Liu et al., 2021), Hessian is proportional to the GNC near local minima when the loss is the negative log likelihood, i.e. cross-entropy loss. To see this, when $w_t \to w^*$, we have $\Sigma_{w^*} = \frac{1}{n}\sum_{i=1}^n \nabla\ell_i\nabla\ell_i^T - G_tG_t^T \approx \frac{1}{n}\sum_{i=1}^n \nabla\ell_i\nabla\ell_i^T = F_{w^*}$, where $F_{w^*}$ is the *Fisher information matrix* (FIM). This approximation is true because gradient noise dominates over gradient mean near local minima. Moreover, FIM is close to the Hessian near local minima with the log-loss (Pawitan, 2001, Chapter 8), namely, $F_{w^*} \approx H_{w^*}$. Let $n \gg b$, we have $H_{w^*} \approx \Sigma_{w^*} = bC_{w^*}$. Consequently, when $\Sigma_T$ is sufficiently close to $\Sigma_{w^*}$, condition (ii) is satisfied. It's important to note that the debate surrounding $H_{w^*} \approx F_{w^*}$ arises when the loss function deviates from cross-entropy (Ziyin et al., 2022).

For condition (iii), the initial learning rate is typically set at a high value, and this condition may not be satisfied until the learning rate undergoes decay in the later stages of SGD training. This observation is evident in Figure 4a-4b, where the condition becomes easily met at the terminal state following the learning rate decay. Moreover, the interplay between $\frac{2}{\eta}$ and $\lambda_1$ is extensively explored in the context of the *edge of stability* (Wu et al., 2018; Cohen et al., 2021; Arora et al., 2022), which suggests that during the training of GD, $\lambda_1$ approaches $\frac{2}{\eta}$ and hovers just above it in the "edge of stability" regime. In the context of Theorem 4.1, as indicated by Lemma 4.1, the diagonal elements of $\Lambda_{w_s^*}$ tend to be close to zero before reaching the "edge of stability" regime, the bound presented in Theorem 4.1 diverges to infinity. This, as a by-product, provides an alternative explanation to the failure mode of $I(W; S) \to \infty$ in the deterministic case, e.g., GD with a fixed initialization.

The following results can be obtained by combining Theorem 4.1 and Lemma 4.1.

**Corollary 4.1.** *Under (i,iii) in Lemma 4.1, then $\mathcal{E}_\mu(\mathcal{A}) \leq \frac{R}{\sqrt{nn}}\sqrt{\mathbb{E}\left[tr\left\{\log\left(\left[H_{w^*}C_T^{-1}\right]\Lambda_{w_\mu^*}\right)\right\}\right]}.$*

(a) VGG on (small) SVHN  (b) VGG on CIFAR10  (c) VGG on (small) SVHN  (d) VGG on CIFAR10

Figure 4: (a-b) The dynamics of $\eta/2 - \lambda_1$. Note that learning rate decays by 0.1 at the $40,000^{\text{th}}$ and the $60,000^{\text{th}}$ iteration. (c-d) The distance of current model parameters from its initialization.

**Corollary 4.2.** *Under (i-iii) in Lemma 4.1, then* $\mathcal{E}_\mu(\mathcal{A}) \leq \sqrt{\frac{dR^2}{n} \log\left(\frac{2b}{\eta d}\mathbb{E}||W_S^* - w_\mu^*||^2 + 1\right)}$.

By $\log(x+1) \leq x$, the bound in Corollary 4.2 is dimension-independent if the weight norm does not grow with $d$. Furthermore, the information-theoretic bound becomes a norm-based bound in Corollary 4.2, which is widely studied in the generalization literature (Bartlett et al., 2017; Neyshabur et al., 2018). In fact, $w_\mu^*$ can be replaced by any data-independent vector, for example, the initialization, $w_0$ (see Corollary E.1). In this case, the corresponding bound suggests that generalization performance can be characterized by the "distance from initialization", namely, given that SGD achieves satisfactory performance on the training data, a shorter distance from the initialization tends to yield better generalization. Nagarajan & Kolter (2019a) also derived a "distance from initialization" based generalization bound by using Rademacher complexity, and Hu et al. (2020) use "distance from initialization" as a regularizer to improve the generalization performance on noisy data.

In the sequel, we use the data-dependent prior bound, namely, Lemma 2.2, to derive new results.

Let $P_{W_T|S_J=s_j} = \mathcal{N}(W_{s_j}^*, \Lambda(W_{s_j}^*))$ where $W_{s_j}^*$ is the local minimum found by the LOO training.

**Theorem 4.2.** *Under the same conditions in Lemma 2.2 and (i-iii) in Lemma 4.1, assuming $\Lambda(W_{s_j}^*)$ is close to $\Lambda(W_s^*)$ for a given $s$, then* $\mathcal{E}_\mu(\mathcal{A}) \leq \mathbb{E}_{S,J}\sqrt{\frac{M^2 b}{2\eta}\mathbb{E}_{W_S^*, W_{S_J}^*}^{S,J}||W_S^* - W_{S_J}^*||^2}$.

This bound implies a strong connection between generalization and the algorithmic stability exhibited by SGD. Specifically, if the hypothesis output does not change much (in the squared $L_2$ distance sense) upon the removal of a single training instance, the algorithm is likely to generalize effectively. In fact, $\mathbb{E}_{W_S^*, W_{S_J}^*}^{S,J}||W_S^* - W_{S_J}^*||^2$ can be regarded as an average version of squared *argument stability* (Liu et al., 2017). Moreover, stability-based bounds often demonstrate a fast decay rate in the convex learning cases (Hardt et al., 2016; Bassily et al., 2020). It is worth noting that if argument stability achieves the fast rate, e.g., $\sup_{s,j}||w_s^* - w_{s_j}^*|| \leq \mathcal{O}(1/n)$, then Theorem 4.2 can also achieve the same rate. In addition, note that the stability-based bound usually contains a Lipshitz constant, while the bound in Theorem 4.2 discards such undesired constant.

Ideally, to estimate the distance of $||w_s^* - w_{s_j}^*||^2$, one can utilize the influence function (Hampel, 1974; Cook & Weisberg, 1982; Koh & Liang, 2017), namely $w_{s_j}^* - w_s^* \approx \frac{1}{n}H_{W_s^*}^{-1}\nabla\ell(w_s^*, z_i)$, where $i$ is the instance index that is not selected in $j$. However, for deep neural network training, the approximation made by influence function is often erroneous (Basu et al., 2021). While this presents a challenge, it motivates further exploration and refinement, seeking to enhance the practical application of Theorem 4.2 in the context of deep learning.

## 5 EMPIRICAL STUDY

In this section, we present some empirical results including tracking training dynamics of SGD and SDE, along with the estimation of several obtained generalization bounds.

**SGD and SDE Training Dynamics** We implement the SDE training by following the same algorithm given in (Wu et al., 2020, Algorithm 1). Our experiments involved training a VGG-11 architecture without BatchNormalization on a subset of SVHN (containing 25k training images) and CIFAR10. Additionally, we trained a ResNet-18 on both CIFAR10 and CIFAR100. Data augmentation is only used in the experiments related to CIFAR100. We ran each experiment for ten

different random seed, maintaining a fixed initialization of the model parameters. Further details about the experimental setup can be found in the Appendix. The results are depicted in Figure 1. As mentioned earlier, SDE exhibits a performance dynamics akin to that of SGD, reinforcing the similarities in their training behaviors.

**Evolution of Key Quantities for SGD and SDE**   We show $||G_t||^2$ and $tr\left\{\log\left(\Sigma_t^{-1}\Sigma_\mu\right)\right\}$ in Figure 2. Recognizing the computational challenges associated with computing $tr\left\{\log\left(\Sigma_t^{-1}\Sigma_\mu\right)\right\}$, we opted to draw estimates based on 100 training and 100 testing samples. Notably, both SGD and SDE exhibit similar behaviors in these gradient-based metrics. It is noteworthy that despite the absence of the learning rate in the trajectories-based bounds, we observed that modifications to the learning rate at the $40,000^{\text{th}}$ and $60,000^{\text{th}}$ steps had discernible effects on these gradient-based quantities. Additionally, in Figure 3, we examine the trace of the Hessian and its largest eigenvalue during training, leveraging the PyHessian library (Yao et al., 2020). Note that we still use only 100 training data to estimate the Hession for efficiency. Notice that the Hessian-related quantities of SGD and SDE are nearly perfectly matched in the terminal state of training. Furthermore, Figures 4c-4d illustrate the "distance to initialization," revealing a consistent trend shared by both SGD and SDE.

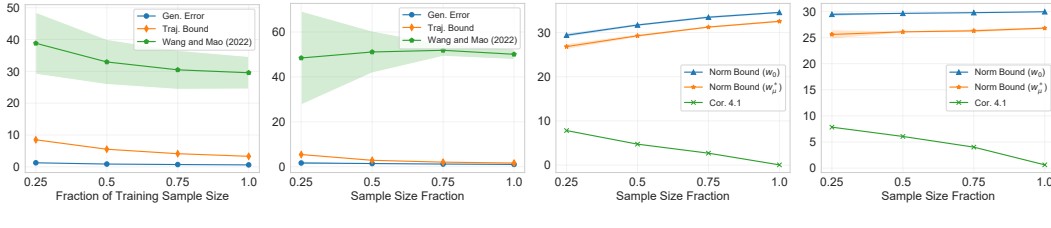

(a) VGG on (small) SVHN     (b) VGG on CIFAR10     (c) VGG on (small) SVHN     (d) VGG on CIFAR10

Figure 5: Estimated trajectories-based bound and terminal-state based bound, with $R$ excluded. Zoomed-in figures of generalization error are given in Figure 7 in Appendix.

**Bound Comparison**   We vary the size of the training sample and empirically estimate several of our bounds in Figure 5, with the subgaussian variance proxy $R$ excluded for simplicity. Thus, the estimated values in Figure 5 don't accurately represent the true order of the bounds. Despite the general unbounded nature of cross-entropy loss, common training strategies, such as proper weight initialization, training techniques, and appropriate learning rate selection, ensure that the cross-entropy loss remains bounded in practice. Therefore, it is reasonable to assume subgaussian behavior of the cross-entropy loss under SGD training. In Figure 5a-5b, we compare our Theorem 3.2 with Wang & Mao (2022, Theorem 2). Since both bounds incorporate the same $R$, the results in Figures 5a to 5b show that our Theorem 3.2 outperforms Wang & Mao (2022, Theorem 2). This aligns with expectations, considering that the isotropic Gaussian used in the auxiliary weight process of Wang & Mao (2022, Theorem 2) is suboptimal, as demonstrated in Lemma 3.2. Moreover, Figures 5c to 5d hint that norm-based bounds Corollary 4.2 and Corollary E.1) exhibit growth with $n$, which are also observed in Nagarajan & Kolter (2019b). In contrast, Corollary 4.1 effectively captures the trend of generalization error, emphasizing the significance of the geometric properties of local minima. Additionally, while trajectories-based bounds may appear tighter, terminal-state-based bounds seem to have a faster decay rate.

## 6   LIMITATIONS AND FUTURE WORKS

While our current work exhibits certain limitations, such as the requirement of positive definiteness for $C_t$ in our trajectories-based bounds, it's worth noting that recent studies (Frankle & Carbin, 2019; Li et al., 2018; Gur-Ari et al., 2018; Larsen et al., 2022) indicate that many parameters in deep neural networks might be dispensable without affecting generalization. This implies that GD/SGD could potentially occur in a subspace of $\mathbb{R}^d$ termed the "intrinsic dimension" $d_{\text{int}}$. Defining $C_t$ within this invertible subspace, utilizing $d_{\text{int}}$, could potentially overcome our current limitations. Theoretical characterization of intrinsic dimension, however, remains an open problem, and further exploration in this direction is poised to significantly improve our work. In addition, extending the similar analytical approach used in this work to other optimizer (e.g., Adam, Adagrad, ...) holds promise.

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

Table 1: Comparison of the results in this work

| | Bounds | Remarks |
|---|---|---|
| | Trajectories-based Bounds. Pros: less assumptions, can track training dynamics; Cro: Time-Dependent | |
| Theorem 3.1 | $\mathcal{O}\left(\sqrt{\frac{d}{n}\mathbb{E}\left[\log\frac{h_1}{d}-\frac{h_2}{d}\right]}\right)$ | Isotropic covariance for Gaussian prior |
| Corollary 3.1 | $\mathcal{O}\left(\sqrt{\frac{d}{n}\sum_{t=1}^{T}\mathbb{E}\log\left(\frac{\mathbb{E}\|G_t-\tilde{g}_t\|^2}{d}+1\right)}\right)$ | Bound for langevin dynamic; tighter than Neu et al. (2021, Prop. 3.) |
| Theorem 3.2 | $\mathcal{O}\left(\sqrt{\frac{1}{n}\sum_{t=1}^{T}\mathbb{E}\left[tr\left\{\log\frac{\Sigma_t^\mu C_t^{-1}}{b}\right\}\right]}\right)$ | Population GNC for prior; tighter than Thm. 3.1 |
| | Terminal-State-based Bounds. Pro: time-indepedent; Cro: more assumptions, cannot track training dynamics | |
| Theorem 4.1 | $\mathcal{O}\left(\sqrt{\frac{1}{n}\mathbb{E}\left[tr\left\{\log\left(\Lambda_{W_S^*}^{-1}\Lambda_{w_\mu^*}\right)\right\}\right]}\right)$ | General result; hard to measure in practice |
| Corollary 4.1 | $\mathcal{O}\left(\sqrt{\frac{1}{n\eta}\mathbb{E}\left[tr\left\{\log\left(\left[H_{w^*}C_T^{-1}\right]\Lambda_{w_\mu^*}\right)\right\}\right]}\right)$ | Under conditions: $H_{w^*}\Lambda_{w^*}=\Lambda_{w^*}H_{w^*}$ and $H_{w^*}\Sigma_T=\mathrm{I}_d$ |
| Corollary 4.2 | $\mathcal{O}\left(\sqrt{\frac{d}{n}\log\left(\frac{b}{\eta d}\mathbb{E}\|W_S^*-\hat{w}\|^2+1\right)}\right)$ | $\hat{w}$ is flexible; $\frac{2}{\eta}\gg\lambda_1$; other conditions same as Cor. 4.1 |
| Theorem 4.2 | $\mathcal{O}\left(\mathbb{E}\sqrt{\frac{M^2b}{\eta}\mathbb{E}\|W_S^*-W_{S_J}^*\|^2}\right)$ | Bounded loss; $\Lambda(W_{s_j}^*)=\Lambda(W_s^*)$; other conditions same as Cor. 4.2 |

# A NOTATION

The distribution of a random variable $X$ is denoted by $P_X$ (or $Q_X$), and the conditional distribution of $X$ given $Y$ is denoted by $P_{X|Y}$. When conditioning on a specific realization $y$, we use the short-hand $P_{X|Y=y}$ or simply $P_{X|Y}$. Denote by $\mathbb{E}_X$ expectation over $X \sim P_X$, and by $\mathbb{E}_{X|Y=y}$ (or $\mathbb{E}_X^y$) expectation over $X \sim P_{X|Y=y}$. We may omit the subscript of the expectation when there is no ambiguity. The KL divergence of probability distribution $Q$ with respect to $P$ is denoted by $\mathrm{D}_{\mathrm{KL}}(Q||P)$. The mutual information (MI) between random variables $X$ and $Y$ is denoted by $I(X;Y)$, and the conditional mutual information between $X$ and $Y$ given $Z$ is denoted by $I(X;Y|Z)$. In addition, for a matrix $A \in \mathbb{R}^{d \times d}$, we let $tr\{A\}$ denote the trace of $A$ and we use $tr\{\log A\}$ to indicate $\sum_{k=1}^{d}\log A_{k,k}$.

# B ADDITIONAL BACKGROUND

## B.1 INFORMATION-THEORETIC BOUNDS FOR SGD

Recently, (Neu et al., 2021; Wang & Mao, 2022) apply information-theoretic analysis to the generalization of models trained with SGD by invoking an auxiliary weight process (AWP). We now denote this auxiliary weight process by $\mathcal{A}_{AWP}$. Let $\mathcal{A}_{SGD}$ be the original algorithm of SGD, (Neu et al., 2021; Wang & Mao, 2022) obtain generalization bounds by the following construction,

$$\mathcal{E}_\mu\left(\mathcal{A}_{SGD}\right)=\mathcal{E}_\mu\left(\mathcal{A}_{SGD}\right)+\mathcal{E}_\mu\left(\mathcal{A}_{AWP}\right)-\mathcal{E}_\mu\left(\mathcal{A}_{AWP}\right)$$

$$\leq\underbrace{\mathcal{O}\left(\sqrt{\frac{I(W_{\mathrm{AWP}};S)}{n}}\right)}_{\text{Lemma 2.1}}+\underbrace{\left|\mathcal{E}_\mu\left(\mathcal{A}_{SGD}\right)-\mathcal{E}_\mu\left(\mathcal{A}_{AWP}\right)\right|}_{\text{residual term}}, \quad (8)$$

where $W_{\mathrm{AWP}}$ is the output hypothesis by $\mathcal{A}_{AWP}$.

Notably, it remains uncertain whether the residual term is sufficiently small for the information-theoretic bounds of $\mathcal{A}_{AWP}$ to yield meaningful insights into SGD. Although there exists an optimal $\mathcal{A}_{AWP}$ that tightens the bound in Eq. (8), finding such an optimal $\mathcal{A}_{AWP}$ beyond the isotropic Gaussian noise covariance case is challenging. It's worth noting that Wang & Mao (2022) provides an optimal bound for the isotropic Gaussian noise case. Nevertheless, our empirical results, as illustrated in Figure 5, demonstrate that the bounds presented in this paper outperform the isotropic Gaussian noise case.

In this paper, we do not attempt to find an optimal $\mathcal{A}_{AWP}$, but instead, we invoke the SDE approximation (i.e. Eq. (5)), denoted as the $\mathcal{A}_{SDE}$. Formally,

$$
\begin{aligned}
\mathcal{E}_\mu\left(\mathcal{A}_{SGD}\right) =& \mathcal{E}_\mu\left(\mathcal{A}_{SGD}\right) + \mathcal{E}_\mu\left(\mathcal{A}_{SDE}\right) - \mathcal{E}_\mu\left(\mathcal{A}_{SDE}\right) \\
\leq & \underbrace{\mathcal{O}\left(\sqrt{\frac{I(W_{\mathrm{SDE}};S)}{n}}\right)}_{\text{Lemma 2.1}} + \underbrace{\left|\mathcal{E}_\mu\left(\mathcal{A}_{SGD}\right) - \mathcal{E}_\mu\left(\mathcal{A}_{SDE}\right)\right|}_{\text{residual term}},
\end{aligned}
\tag{9}
$$

where $W_{\mathrm{SDE}}$ is the output hypothesis by $\mathcal{A}_{SDE}$.

Empirical evidence from (Wu et al., 2020; Li et al., 2021) and our Figure 1 suggests that the residual term in Eq. (9) is small. This observation motivates our investigation into the generalization of SGD using the information-theoretic bounds of SDE directly.

### B.2 Theoretical Validation of SDE

To theoretically assess the validation of SDE in approximating SGD, two essential technical definitions are necessary.

**Definition B.1** (Test Functions). *Class $\mathcal{F}$ of continuous functions $\mathbb{R}^d \to \mathbb{R}$ has polynomial growth if $\forall\ f \in \mathcal{F}$, if there exists constants $K, \kappa > 0$ s.t. $|f(x)| < K(1 + |x|^\kappa)$ for all $x \in \mathbb{R}$.*

**Definition B.2** (Order$-\alpha$ weak approximation). *Let $\eta \in (0,1)$, $T > 0$ and $N = \lfloor T/\eta \rfloor$. Let $\mathcal{F}$ be the set of test Functions. We say that the SDE in Eq. (4) is an order $\alpha$ weak approximation of the SGD in Eq. (1) if for every $f \in \mathcal{F}$, there exists $C > 0$, independent of $\eta$, s.t. for all $k = 0, 1, \ldots, N$,*

$$
\left|\mathbb{E}\left[f(\omega_{k\eta})\right] - \mathbb{E}\left[f(W_k)\right]\right| \leq C\eta^\alpha.
$$

Below is a classical result.

**Lemma B.1** (Li et al. (2017, Theorem 1)). *Assume $\nabla\ell$ is Lipschitz continuous, has at most linear asymptotic growth and has sufficiently high derivatives belonging to $\mathcal{F}$, then SDE in Eq. (4) is an order $1$ weak approximation of the SGD in Eq. (1). Or equivalently, for every $f \in \mathcal{F}$, there exists $C > 0$, independent of $\eta$, s.t. $\max_{k=0,1,\ldots,N}\left|\mathbb{E}\left[f(\omega_{k\eta})\right] - \mathbb{E}\left[f(W_k)\right]\right| \leq C\eta$.*

This theorem suggests that SGD and SDE closely track each other when they result in similar distributions of outcomes, such as the returned hypothesis $W$. In addition, the closeness of distributions is formulated through expectations of suitable classes of test functions, as defined in Definition B.1. As mentioned in Li et al. (2021), of particular interest for machine learning are test functions like generalization error $\mathcal{E}_\mu$, which may not adhere to formal conditions such as differentiability assumed in classical theory but are still valuable for experimental use. Other typical choices of test functions includes weight norm, gradient norm, and the trace of noise covariance.

### B.3 Gaussian Distribution around Local Minimum

A multi-dimensional Ornstein-Uhlenbeck process is defined as

$$
dx_t = -\mathbf{H}x_t dt + \mathbf{B}d\theta_t,
\tag{10}
$$

where $x_t \in \mathbb{R}^d$, $\mathbf{H}$, $\mathbf{B}$ are $d \times d$ matrices and $\theta_t$ is an $d$-dimensional Wiener process.

Denote the density function of $x_t$ as $P(x,t)$, then the corresponding Fokker-Planck equation describes the evolution of $P(x,t)$:

$$
\frac{\partial P(x,t)}{\partial t} = \sum_{i=1}^{d}\sum_{j=1}^{d}\frac{\partial}{\partial x_i}\left(P(x,t)\sum_{j=1}^{d}\mathbf{H}_{i,j}x_j\right) + \sum_{i=1}^{d}\sum_{j=1}^{d}\mathbf{D}_{i,j}\frac{\partial^2 P(x,t)}{\partial x_i \partial x_j},
$$

where $\mathbf{D} = \mathbf{B}\mathbf{B}^{\mathbf{T}}/2$.

Moreover, if $\mathbf{H}$ is positive define, then a stationary solution of $P$ is given by (Freidlin & Wentzell, 2012):

$$
P(x) = \frac{1}{\sqrt{(2\pi)^d |\Sigma|}}\exp\left(-\frac{1}{2}x^{\mathbf{T}}\Sigma^{-1}x\right),
\tag{11}
$$

where $\Sigma = \mathbb{E}\left[xx^{\mathbf{T}}\right]$ is the covariance matrix of $x$.

When $w$ is close to any local minimum $w^*$, we can use a second-order Taylor expansion to approximate the value of the loss at $w$,

$$L_s(w) \approx L_s(w^*) + \frac{1}{2}(w - w^*)^{\mathbf{T}} H_{w^*}(w - w^*). \tag{12}$$

In this case, when $w_t \to w^*$, we have $G_t = \nabla L_s(w_t) = H_{w^*}(w_t - w^*)$. Recall Eq. (2), then

$$w_t = w_{t-1} - \eta G_t + \eta V_t = w_{t-1} - \eta H_{w^*}(w_{t-1} - w^*) + \eta V_t.$$

Let $W_t' \triangleq W_t - w^*$ and recall Eq (10), we thus have the Ornstein-Uhlenbeck process for $x_t = W_t'$ as

$$dW_t' = -\eta H_{w^*} W_t' dt + \eta \sqrt{C_t} d\theta_t. \tag{13}$$

By Eq. (11), we have

$$P(W') \propto \exp\left(-\frac{1}{2}W'^{\mathbf{T}}\Lambda_{w^*}^{-1}W'\right).$$

Consequently, the stationary distribution of $W$ for a given $w^*$ is $\mathcal{N}(w^*, \Lambda_{w^*})$.

For discrete case, we have

$$
\begin{aligned}
w_t' &= \left(\mathrm{I}_d - \eta H_{w^*}\right) w_{t-1}' + \eta V_t \\
&= \left(\mathrm{I}_d - \eta H_{w^*}\right)^2 w_{t-2}' + \eta\left(\left(\mathrm{I}_d - \eta H_{w^*}\right) V_{t-1} + V_t\right) \\
&\vdots \\
&= \bar{H}^t w_0' + \eta \sum_{i=0}^{t} \bar{H}^i V_{t-i},
\end{aligned}
$$

where $\bar{H} = \mathrm{I}_d - \eta H_{w^*}$. Notably, when $t$ is sufficiently large, then the first term is negligible, especially with a small learning rate, we have $w_t' = w_t - w^* = \eta \sum_{i=0}^{t} \bar{H}^i V_{t-i}$. When $C_t$ does not change in the long time limit, then $W_t'$ is the weighted sum of independent Gaussian random variables, which follows a Gaussian distribution, namely $w_t \sim \mathcal{N}(w^*, \Lambda_{w^*})$. We refer readers to (Liu et al., 2021, Theorem 1-2.) for a relaxed analysis in the discrete case.

## C  SOME USEFUL LEMMAS

We present the variational representation of mutual information below.

**Lemma C.1** (Polyanskiy & Wu (2019, Corollary 3.1.)). *For two random variables $X$ and $Y$, we have*

$$I(X;Y) = \inf_P \mathbb{E}_X\left[\mathrm{D}_{\mathrm{KL}}(Q_{Y|X}\|P)\right],$$

*where the infimum is achieved at $P = Q_Y$.*

The following lemma is inspired by the classic Log sum inequality in Cover & Thomas (2012, Theorem 2.7.1).

**Lemma C.2.** *For non-negative numbers $\{a_i\}_{i=1}^n$ and $\{b_i\}_{i=1}^n$,*

$$\sum_{i=1}^n b_i \log \frac{a_i}{b_i} \leq \left(\sum_{i=1}^n b_i\right) \log \frac{\sum_{i=1}^n a_i}{\sum_{i=1}^n b_i},$$

*with equality if and only if $\frac{a_i}{b_i} = const.$*

*Proof.* Since $\log$ is a concave function, according to Jensen's inequality, we have

$$\sum_{i=1}^n \alpha_i \log(x_i) \leq \log(\sum_{i=1}^n \alpha_i x_i),$$

where $\sum_{i=1}^{n} \alpha_i = 1$.

Let $\alpha_i = \frac{b_i}{\sum_{i=1}^{n} b_i}$ and $x_i = \frac{a_i}{b_i}$, and plugging them into the inequality above, we have

$$\sum_{i=1}^{n} \frac{b_i}{\sum_{i=1}^{n} b_i} \log(\frac{a_i}{b_i}) \leq \log \left( \sum_{i=1}^{n} \frac{b_i}{\sum_{i=1}^{n} b_i} \frac{a_i}{b_i} \right) = \log \left( \frac{\sum_{i=1}^{n} a_i}{\sum_{i=1}^{n} b_i} \right),$$

which implies

$$\sum_{i=1}^{n} b_i \log(\frac{a_i}{b_i}) \leq \left( \sum_{i=1}^{n} b_i \right) \log \left( \frac{\sum_{i=1}^{n} a_i}{\sum_{i=1}^{n} b_i} \right).$$

This completes the proof. $\qquad\square$

Below is the KL divergence between two Gaussian distributions $p = \mathcal{N}(\mu_p, \Sigma_p)$ and $q = \mathcal{N}(\mu_q, \Sigma_q)$, where $\mu_p, \mu_q \in \mathbb{R}^d$ and $\Sigma_p, \Sigma_q \in \mathbb{R}^{d \times d}$.

$$\mathrm{D_{KL}}(p||q) = \frac{1}{2} \left[ \log \frac{\det(\Sigma_q)}{\det(\Sigma_p)} - d + (\mu_p - \mu_q)^T \Sigma_q^{-1} (\mu_p - \mu_q) + tr \left\{ \Sigma_q^{-1} \Sigma_p \right\} \right]. \qquad (14)$$

## D    OMITTED PROOFS AND ADDITIONAL RESULTS IN SECTION 3

### D.1    LEMMA D.1: UNROLLING MUTUAL INFORMATION

We first unroll the terminal parameters' mutual information $I(W_T; S)$ to the full trajectories' mutual information via the lemma below.

**Lemma D.1.** $I(W_T; S) \leq \sum_{t=1}^{T} I(-G_t + C_t^{1/2} N_t; S|W_{t-1})$.

This lemma can be proved by recurrently applying the data processing inequality (DPI) and chain rule of the mutual information (Polyanskiy & Wu, 2019).

*Proof.* Recall the SDE approximation of SGD, i.e., Eq (5), we then have,

$$
\begin{aligned}
I(W_T; S) &= I(W_{T-1} - \eta G_T + \eta C_T^{1/2} N_T; S) \\
&\leq I(W_{T-1}, -\eta G_T + \eta C_T^{1/2} N_T; S) & (15) \\
&= I(W_{T-1}; S) + I(-\eta G_T + \eta C_T^{1/2} N_T; S|W_{T-1}) & (16) \\
&\vdots \\
&\leq \sum_{t=1}^{T} I(-\eta G_t + \eta C_t^{1/2} N_t; S|W_{t-1}) \\
&= \sum_{t=1}^{T} I(-G_t + C_t^{1/2} N_t; S|W_{t-1}).
\end{aligned}
$$

where Eq. (15) is by the data processing inequality (e.g., $Z - (X, Y) - (X + Y)$ form a markov chain then $I(X + Y, Z) \leq I(X, Y; Z)$), Eq. (16) is by the chain rule of the mutual information, and learning rate $\eta$ is dropped since mutual information is scale-invariant. $\qquad\square$

### D.2    PROOF OF LEMMA 3.1

*Proof.* For any $t \in [T]$, similar to the proof of Lemma C.1 in Polyanskiy & Wu (2019):

$$
\begin{aligned}
&I(-G_t + C_t^{1/2} N_t; S|W_{t-1} = w_{t-1}) \\
&= \mathbb{E}_S^{w_{t-1}} \left[ \mathrm{D_{KL}}(Q_{\widehat{G}_t|w_{t-1},S} || Q_{\widehat{G}_t|w_{t-1}}) \right] \\
&= \mathbb{E}_S^{w_{t-1}} \left[ \mathrm{D_{KL}}(Q_{\widehat{G}_t|w_{t-1},S} || P_{\widehat{G}_t|w_{t-1}}) - \mathrm{D_{KL}}(Q_{\widehat{G}_t|w_{t-1}} || P_{\widehat{G}_t|w_{t-1}}) \right] \\
&\leq \mathbb{E}_S^{w_{t-1}} \left[ \mathrm{D_{KL}}(Q_{\widehat{G}_t|w_{t-1},S} || P_{\widehat{G}_t|w_{t-1}}) \right], & (17)
\end{aligned}
$$

where Eq. (17) is due to the fact that KL divergence is non-negative, and the equality holds when $P_{\widehat{G}_t|w_{t-1}} = Q_{\widehat{G}_t|w_{t-1}}$ for $W_{t-1} = w_{t-1}$.

Thus, we conclude that

$$I(\widehat{G}_t; S|W_{t-1} = w_{t-1}) = \inf_{P_{\widehat{G}_t|w_{t-1}}} \mathbb{E}_S^{w_{t-1}} \left[ D_{\mathrm{KL}}(Q_{\widehat{G}_t|w_{t-1},S}||P_{\widehat{G}_t|w_{t-1}}) \right].$$

Taking expectation over $W_{t-1}$ for both side above, we have

$$I(\widehat{G}_t; S|W_{t-1}) = \mathbb{E}_{W_{t-1}} \left[ \inf_{P_{\widehat{G}_t|W_{t-1}}} \mathbb{E}_S^{W_{t-1}} \left[ D_{\mathrm{KL}}(Q_{\widehat{G}_t|W_{t-1},S}||P_{\widehat{G}_t|W_{t-1}}) \right] \right].$$

This completes the proof. $\qquad\square$

### D.3 PROOF OF THEOREM 3.1

*Proof.* We first prove Eq. (6). Recall Lemma 3.1 and assume $C_t$ is a positive-definite matrix, for any $t \in [T]$, we have

$$I(-G_t + C_t^{1/2} N_t; S|W_{t-1} = w_{t-1})$$

$$\leq \inf_{\tilde{g}_t, \sigma_t} \mathbb{E}_S^{w_{t-1}} \left[ D_{\mathrm{KL}}(Q_{-G_t+C_t^{1/2}N_t|w_{t-1},S}||P_{-\tilde{g}_t+\sigma_t N_t|w_{t-1}}) \right]$$

$$= \inf_{\tilde{g}_t, \sigma_t} \mathbb{E}_S^{w_{t-1}} \left[ \frac{1}{2} \left[ \log \frac{\det(\sigma_t^2 \mathrm{I}_d)}{\det(C_t)} - d + \frac{1}{\sigma_t^2}((G_t - \tilde{g}_t)^T \mathrm{I}_d^{-1}(G_t - \tilde{g}_t)) + \frac{1}{\sigma_t^2} tr\left\{ \mathrm{I}_d^{-1} C_t \right\} \right] \right] \quad (18)$$

$$= \frac{1}{2} \inf_{\tilde{g}_t, \sigma_t} \mathbb{E}_S^{w_{t-1}} \left[ \frac{1}{\sigma_t^2} \left( ||G_t - \tilde{g}_t||^2 + tr\left\{ C_t \right\} \right) + d \log \sigma_t^2 - d - tr\left\{ \log C_t \right\} \right], \quad (19)$$

where Eq. (18) is by Eq. (14), Eq. (19) is due to the fact that $\log \det(C_t) = tr\{\log C_t\}$ when $C_t$ is positive definite.

Recall that $h_1(w) = \mathbb{E}_S^w \left[ ||G_t - \tilde{g}_t||^2 + tr\{C_t\} \right]$ and $h_2(w) = \mathbb{E}_S^w \left[ tr\{\log C_t\} \right]$, then we have

$$\frac{1}{2} \inf_{\tilde{g}_t, \sigma_t} \frac{1}{\sigma_t^2} \mathbb{E}_S^{w_{t-1}} \left[ ||G_t - \tilde{g}_t||^2 + tr\{C_t\} \right] + d \log \sigma_t^2 - d - \mathbb{E}_S^{w_{t-1}} \left[ tr\{\log C_t\} \right]$$

$$\leq \frac{1}{2} \inf_{\sigma_t > 0} \frac{1}{\sigma_t^2} h_1(w_{t-1}) + d \log \sigma_t^2 - d - h_2(w_{t-1})$$

$$= \frac{1}{2} d \log \frac{h_1(w_{t-1})}{d} - \frac{1}{2} h_2(w_{t-1}),$$

where we fix an arbitrary $\tilde{g}_t$ and use the optimal $\sigma^* = \sqrt{\frac{h_1(w_{t-1})}{d}}$.

Plugging everything into Lemma D.1 and Lemma 2.1 will obtain Eq. (6).

We then prove the second part. Let $\tilde{g}_t = \mathbb{E}_Z \left[ \nabla \ell(w_{t-1}, Z) \right]$, then

$$h_1(W_{t-1}) = \mathbb{E}_S^{W_{t-1}} \left[ ||G_t - \tilde{g}_t||^2 + tr\{C_t\} \right]$$

$$= \mathbb{E}_S^{W_{t-1}} \left[ tr\left\{ (G_t - \tilde{g}_t)((G_t - \tilde{g}_t)^T) \right\} \right] + tr\left\{ \mathbb{E}_S^{W_{t-1}} [C_t] \right\}$$

$$= \frac{1}{n} tr\{\Sigma_t^\mu\} + \frac{n-b}{b(n-1)} tr\left\{ \mathbb{E}_S^{W_{t-1}} [\Sigma_t] \right\} \quad (20)$$

$$= \frac{1}{n} tr\{\Sigma_t^\mu\} + \frac{n-b}{bn} tr\{\Sigma_t^\mu\} \quad (21)$$

$$= \frac{1}{b} tr\{\Sigma_t^\mu\},$$

where Eq. (20) is by $\mathbb{E}_S \left[ (G_t - \tilde{g}_t)((G_t - \tilde{g}_t)^T) \right] = \frac{1}{n} \Sigma_t^\mu$ for a given $W_{t-1} = w_{t-1}$ and $C_t = \frac{n-b}{b(n-1)} \Sigma_t$, and Eq. (21) is by $\mathbb{E}_S [\Sigma_t] = \frac{n-1}{n} \Sigma_t^\mu$. This completes the proof. $\qquad\square$

**Remark D.1.** *Unlike information-theoretic generalization bounds of SGLD in the literature, learning rate does not explicitly appear in the bound of Theorem 3.1. This is because the noise random variable has a tunable scaling factor in SGLD while the noise random variable has a fixed scaling factor $\eta$. The latter scaling factor is then dropped since mutual information is scale-invariant (see the proof of Lemma D.1 for more details).*

## D.4 Proof of Corollary 3.1

*Proof.* Let $C_t = \mathrm{I}_d$, by Theorem 3.1,

$$
\begin{aligned}
\mathcal{E}_\mu(\mathcal{A}) &\leq \sqrt{\frac{R^2}{n} \sum_{t=1}^{T} d\mathbb{E}_{W_{t-1}} \left[ \log \frac{\mathbb{E}_S^{W_{t-1}} \left[ ||G_t - \tilde{g}_t||^2 + tr\{C_t\} \right]}{d} \right] - \mathbb{E}_{W_{t-1},S} \left[ tr\{\log C_t\} \right]} \\
&= \sqrt{\frac{R^2}{n} \sum_{t=1}^{T} d\mathbb{E}_{W_{t-1}} \left[ \log \frac{\mathbb{E}_S^{W_{t-1}} \left[ ||G_t - \tilde{g}_t||^2 + d \right]}{d} \right]} \\
&= \sqrt{\frac{R^2}{n} \sum_{t=1}^{T} d\mathbb{E}_{W_{t-1}} \left[ \log \frac{\mathbb{E}_S^{W_{t-1}} \left[ ||G_t - \tilde{g}_t||^2 \right]}{d} + 1 \right]}.
\end{aligned}
$$

This completes the proof. $\qquad\square$

## D.5 Proof of Theorem 3.2

*Proof.* Recall Lemma 3.1, we have

$$
\begin{aligned}
&I(-G_t + C_t^{1/2} N_t; S | W_{t-1} = w_{t-1}) \\
&\leq \inf_{\tilde{c}_t} \mathbb{E}_S^{w_{t-1}} \left[ \mathrm{D_{KL}}(Q_{\widehat{G}_t | w_{t-1}, S} || P_{\widehat{G}_t | w_{t-1}}) \right] \\
&= \inf_{\tilde{c}_t} \mathbb{E}_S^{w_{t-1}} \left[ \frac{1}{2} \left[ \log \frac{\det(\tilde{c}_t \Sigma_t^\mu)}{\det(C_t)} - d + \frac{1}{\tilde{c}_t}((G_t - \tilde{g}_t)^T (\Sigma_t^\mu)^{-1} (G_t - \tilde{g}_t)) + \frac{1}{\tilde{c}_t} tr\left\{ (\Sigma_t^\mu)^{-1} C_t \right\} \right] \right] \\
&= \frac{1}{2} \inf_{\tilde{c}_t} \frac{1}{\tilde{c}_t} tr\left\{ (\Sigma_t^\mu)^{-1} \mathbb{E}_S^{w_{t-1}} \left[ (G_t - \tilde{g}_t)((G_t - \tilde{g}_t)^T] \right\} \right. \\
&\qquad\qquad + \frac{1}{\tilde{c}_t} tr\left\{ (\Sigma_t^\mu)^{-1} \mathbb{E}_S^{w_{t-1}} [C_t] \right\} + tr\left\{ \log \Sigma_t^\mu - \mathbb{E}_S^{w_{t-1}} [\log C_t] \right\} + d\log \tilde{c}_t - d \\
&= \frac{1}{2} \inf_{\tilde{c}_t} \frac{1}{\tilde{c}_t n} tr\left\{ (\Sigma_t^\mu)^{-1} \Sigma_t^\mu \right\} + \frac{n-b}{\tilde{c}_t bn} tr\left\{ (\Sigma_t^\mu)^{-1} \Sigma_t^\mu \right\} + tr\left\{ \log \Sigma_t^\mu - \mathbb{E}_S^{w_{t-1}} [\log C_t] \right\} + d\log \tilde{c}_t - d
\end{aligned}
$$
$$(22)$$
$$
\begin{aligned}
&= \frac{1}{2} \inf_{\tilde{c}_t} \frac{d}{\tilde{c}_t n} + \frac{(n-b)d}{\tilde{c}_t bn} + tr\left\{ \log \Sigma_t^\mu - \mathbb{E}_S^{w_{t-1}} [\log C_t] \right\} + d\log \tilde{c}_t - d \\
&= \frac{1}{2} \inf_{\tilde{c}_t} \frac{d}{b\tilde{c}_t} + d\log \tilde{c}_t + tr\left\{ \log \Sigma_t^\mu - \mathbb{E}_S^{w_{t-1}} [\log C_t] \right\} - d \\
&= \frac{d}{2} \log \frac{1}{b} + \frac{1}{2} tr\left\{ \log \Sigma_t^\mu - \mathbb{E}_S^{w_{t-1}} [\log C_t] \right\},
\end{aligned}
$$

where the last equality hold when $\tilde{c}_t^* = 1/b$ and Eq. (22) is by

$$
\mathbb{E}_S^{w_{t-1}} \left[ (G_t - \tilde{g}_t)((G_t - \tilde{g}_t)^T) \right] = \frac{1}{n} \Sigma_t^\mu, \quad \text{and}
$$

$$
\mathbb{E}_S^{w_{t-1}} [C_t] = \frac{n-b}{b(n-1)} \mathbb{E}_S^{w_{t-1}} [\Sigma_t] = \frac{n-b}{b(n-1)} \frac{n-1}{n} \Sigma_t^\mu = \frac{n-b}{bn} \Sigma_t^\mu.
$$

This completes the proof. $\qquad\square$

### D.6 Proof of Lemma 3.2

*Proof.* Let the diagonal element of $\Sigma_t^\mu/b$ in dimension $k$ be $a_k$, then

$$\sum_{k=1}^d \log a_k \le (\sum_{k=1}^d 1) \cdot \log\left(\sum_{k=1}^d a_k\right)/(\sum_{k=1}^d 1) = d\log(tr\,\{\Sigma_t^\mu\}/bd),$$

where we invoke Lemma C.2.

This completes the proof. $\qquad\square$

### D.7 Additional Result via Data-Dependent Prior

With the same spirit of Lemma D.1, to apply Lemma 2.2 to iterative algorithms, we also need the lemma below, which using the full training trajectories KL divergence to upper bound the final output KL divergence.

**Lemma D.2** (Negrea et al. (2019, Proposition 2.6.))**.** *Assume that $P_{W_0} = Q_{W_0}$, then* $D_{KL}(P_{W_T}||Q_{W_T}) \le \sum_{t=1}^T \mathbb{E}_{W_{0:t-1}} \left[ D_{KL}(P_{W_t|W_{0:t-1}}||Q_{W_t|W_{0:t-1}}) \right].$

Let $G_{Jt} \triangleq \nabla L_{S_J}(W_{t-1})$, the SDE approximation of this prior updating is defined as:

$$W_t = W_{t-1} - \eta G_{Jt} + \eta C_{Jt}^{\frac{1}{2}} N_t,$$

where $C_{Jt} = \frac{1}{b}\left(\frac{1}{m}\sum_{i\in J}\nabla\ell_i\nabla\ell_i^T - G_{Jt}G_{Jt}^T\right)$ is the gradient noise covariance of the prior process. In this case, the prior distribution $P_{\mathcal{G}_{Jt}|W_{0:t-1}}$ will be an anisotropic Gaussian distribution. We also assume $n \gg b$, then $C_t = \frac{1}{b}\Sigma_t$.

We denote the difference between $G_t$ and $G_{Jt}$ by

$$\xi_t \triangleq G_{Jt} - G_t.$$

To see the relationship between $\xi_t$, $C_{Jt}$ and $C_t$, we present a useful lemma below.

**Lemma D.3.** *If $m = n - 1$, then the following two equations hold,*

$$\mathbb{E}\left[\xi_t\xi_t^T\right] = \frac{b}{(n-1)^2}C_t, \quad \mathbb{E}\left[C_{Jt}\right] = \frac{n(n-2)}{(n-1)^2}C_t,$$

*where the expectation is taken over $J$.*

Instead of using Lemma 2.2, we invoke the following result which is a simple extension of (Negrea et al., 2019, Theorem 2.5).

**Lemma D.4** ((Wang et al., 2021a, Theorem 1.))**.** *Assume the loss $\ell(w, Z)$ is bounded in $[0, M]$, the expected generalization gap is bounded by*

$$\mathcal{E}_\mu(\mathcal{A}) \le \frac{M}{\sqrt{2}}\mathbb{E}_{S,J}\sqrt{D_{KL}(P_{W|S_J}||Q_{W|S})}$$

**Comparison with the work of Wang et al. (2021a)**    Wang et al. (2021a) studies the algorithm of SGD with anisotropic noise, while our SDE analysis focuses on GD with anisotropic noise. This means that the discrete gradient noise arising from mini-batch sampling still exists in their analyzed algorithm, whereas the gradient noise is fully modeled as Gaussian in our Section B.1. Moreover, Wang et al. (2021a) uses matrix analysis tools to optimize the prior distribution. A significant distinction lies in their optimization analysis, which relies on the assumption that the trace of gradient noise covariance remains unchanged during training (see **Constraint 1** in their paper). Additionally, their final optimal posterior covariance is derived based on the assumption that the posterior distribution of $W$ is invariant to the data index, see Assumption 1 in their paper. In contrast, our Section B.1 avoids making these assumptions and demonstrates the superiority of population gradient noise covariance (GNC) in Lemma 3.2, by invoking a variant of the log-sum inequality. In summary, our

proof is simpler and more straightforward, while Wang et al. (2021a) makes a stronger claim about the optimality of population GNC based on their additional assumptions.

As introduced in Wang et al. (2021a), the subsequent analysis based on the data-dependent prior bound will rely on an additional assumption.

**Assumption 1.** *When $m = n - 1$, given dataset $S = s$, the distribution $P_{W_t|J,S_J}$ is invariant of $J$.*

In Wang et al. (2021a), authors mention that in practice, $n$ is usually very large, so this assumption hints that changing one instance in $S_J$ will not make $P_{W_t|J,S_J}$ be too different.

We are now in a position to state the following theorem.

**Theorem D.1.** *Assume the loss $\ell(w, Z)$ is bounded in $[0, M]$ and Assumption 1 hold, the expected generalization gap of SGD is bounded by*

$$\mathcal{E}_\mu(\mathcal{A}) \leq \mathbb{E}_S \left[ \sqrt{M^2 \sum_{t=1}^{T} \mathbb{E}_{W_{t-1}} \left[ \left( \frac{(b-1)d}{(n-1)^2} + tr \left\{ \mathbb{E}_J \left[ \log C_t C_{Jt}^{-1} \right] \right\} \right) \right]} \right].$$

*Proof.* By Lemma D.4 and Lemma D.2, we have

$$\mathcal{E}_\mu(\mathcal{A}) \leq \mathbb{E}_{S,J} \left[ \sqrt{\frac{R'^2}{2} \sum_{t=1}^{T} \mathbb{E}_{W_{0:t-1}|S,J} \left[ D_{\mathrm{KL}}(P_{W_t|W_{0:t-1},S_J} \| Q_{W_t|W_{0:t-1},s}) \right]} \right]$$

$$\leq \mathbb{E}_S \left[ \sqrt{\frac{R'^2}{2} \sum_{t=1}^{T} \mathbb{E}_{W_{0:t-1}|S,J} \left[ \mathbb{E}_J \left[ D_{\mathrm{KL}}(P_{W_t|W_{0:t-1},S_J} \| Q_{W_t|W_{0:t-1},s}) \right] \right]} \right], \quad (23)$$

where Eq. (23) is by Jensen's inequality and Assumption 1.

Recall $\Sigma_t = \frac{1}{n} \sum_{i=1}^{n} \nabla \ell(W_{t-1}, Z_i) \nabla \ell(W_{t-1}, Z_i)^T - \nabla L_S(W_{t-1}) \nabla L_S(W_{t-1})^T$ and $C_t = \frac{1}{b} \Sigma_t$.

By the KL divergence between two Gaussian distributions, for any $t \in [T]$, we have

$$\mathbb{E}_J \left[ D_{\mathrm{KL}}(P_{W_t|W_{0:t-1},S_J} \| Q_{W_t|W_{0:t-1},s}) \right]$$

$$= \mathbb{E}_J \left[ \frac{1}{2} \left( \xi_t^T C_t^{-1} \xi_t + \log \frac{\det(C_t)}{\det(C_{Jt})} + tr\{C_t^{-1} C_{Jt}\} - d \right) \right] \quad (24)$$

$$= \frac{1}{2} \left( tr\{C_t^{-1} \mathbb{E}_J \left[ \xi_t \xi_t^T \right]\} + \mathbb{E}_J \left[ \log \frac{\det(C_t)}{\det(C_{Jt})} \right] + \mathbb{E}_J \left[ tr\{C_t^{-1} C_{Jt}\} \right] - d \right)$$

$$= \frac{1}{2} \left( \frac{1}{(n-1)^2} tr\{C_t^{-1} \Sigma_t\} + \mathbb{E}_J \left[ \log \frac{\det(C_t)}{\det(C_{Jt})} \right] + \mathbb{E}_J \left[ tr\{C_t^{-1} C_{Jt}\} \right] - d \right) \quad (25)$$

$$= \frac{1}{2} \left( \frac{b}{(n-1)^2} tr\{\Sigma_t^{-1} \Sigma_t\} + \mathbb{E}_J \left[ \log \frac{\det(C_t)}{\det(C_{Jt})} \right] + tr\{C_t^{-1} \mathbb{E}_J \left[ C_{Jt} \right]\} - d \right)$$

$$= \frac{1}{2} \left( \frac{bd}{(n-1)^2} + \frac{n(n-2)d}{(n-1)^2} - d + tr\{\log C_t - \mathbb{E}_J \left[ \log C_{Jt} \right]\} \right) \quad (26)$$

$$= \frac{1}{2} \left( \frac{(b-1)d}{(n-1)^2} + tr\{\log C_t - \mathbb{E}_J \left[ \log C_{Jt} \right]\} \right)$$

where Eq. (25) and Eq. (26) are by Lemma D.3. This concludes the proof. $\qquad \square$

**Remark D.2.** *If the bound in Negrea et al. (2019) is used, then the first term in Eq. (24) is $\xi_t^{\mathbf{T}} C_{Jt}^{-1} \xi_t$, where both $C_{Jt}$ and $\xi_t$ dependent on $J$, making the bound difficult to analyze.*

The effect of $tr\{\log C_t\}$ on the magnitude of the bound can be decreased by the $tr\{\mathbb{E}_J \log C_{Jt}\}$. If we further consider Taylor expansion of the function $\log C_{Jt}$ around $\mathbb{E}_J[C_{Jt}]$, we have a well-known approximation

$$\mathbb{E} \left[ \log C_{Jt} \right] \approx \log \mathbb{E} \left[ C_{Jt} \right] - \mathrm{Var}(C_{Jt}) / (2\mathbb{E}^2 \left[ C_{Jt} \right]).$$

Thus, recall Lemma D.3, the difference between $tr\{\log C_t\}$ and $tr\{\mathbb{E}_J \log C_{Jt}\}$ would become:

$$\log\left(1 + 1/(n^2 - 2n)\right) + \mathrm{Var}(C_{Jt})/(2\mathbb{E}^2[C_{Jt}]).$$

Thus, the generalization gap should be characterized by the second term above.

When $n \to \infty$, the first term will converges to zero, and for the second term, $\mathbb{E}^2[C_{Jt}]$ will converge to a constant by Lemma D.3, and then the bound is $\mathrm{Var}(C_{Jt})$ will also converges to zero.

## E  OMITTED PROOFS, ADDITIONAL RESULTS AND DISCUSSIONS IN 4

In fact, this section provides a PAC-Bayes type analysis. The connection between information-theoretic bounds and PAC-Bays bounds have already been discussed in many previous works (Bassily et al., 2018; Hellström & Durisi, 2020; Alquier, 2021). Roughly speaking, the most significant component of a PAC-Bayes bound is the KL divergence between the posterior distribution of a randomized algorithm output and a prior distribution, i.e. $\mathrm{D_{KL}}(Q_{W_T|S}||P_N)$ for some prior $P_N$. In essence, information-theoretic bounds can be view as having the same spirit. For concreteness, in Lemma 2.1, $I(W_T; S) = \mathbb{E}_S[\mathrm{D_{KL}}(Q_{W_T|S}||P_{W_T})]$, in which case the marginal $P_{W_T}$ is used as a prior of the algorithm output. Furthermore, by using Lemma C.1, we have $I(W_T; S) \leq \inf_{P_N} \mathbb{E}_S[\mathrm{D_{KL}}(Q_{W_T|S}||P_N)]$. Hence, Lemma 2.1 can be regarded as a PAC-Bayes bound with the optimal prior. In addition, the PAC-Bayes framework is usually used to provide a high-probability bound, while information-theoretic analysis is applied to bounding the expected generalization error. In this sense, information-theoretic framework is closer to another concept called MAC-Bayes (Grunwald et al., 2021).

### E.1  PROOF OF LEMMA 4.1

*Proof.* Recall $G_t = \nabla L_s(w_t) = H_{w^*}(w_t - w^*)$. and Eq. (2), then

$$\begin{aligned} w_t =& w_{t-1} - \eta G_t + \eta V_t \\ =& w_{t-1} - \eta H_{w^*}(w_{t-1} - w^*) + \eta V_t. \end{aligned}$$

Let $W'_t \triangleq W_t - w^*$. Thus, as $T \to \infty$,

$$\mathbb{E}_{W'_T}\left[W'_T W'_T{}^{\mathbf{T}}\right]$$
$$=\mathbb{E}_{W'_{T-1}, V_T}\left[\left(W'_{T-1} - \eta H_{w^*} W'_{T-1} + \eta V_t\right)\left(W'_{T-1} - \eta H_{w^*} W'_{T-1} + \eta V_t\right)^{\mathbf{T}}\right]$$
$$=\mathbb{E}_{W'_{T-1}}\left[W'_{T-1} W'_{T-1}{}^{\mathbf{T}} - \eta H_{w^*} W'_{T-1} W'_{T-1}{}^{\mathbf{T}} - \eta W'_{T-1} W'_{T-1}{}^{\mathbf{T}} H_{w^*} + \eta^2 H_{w^*} W'_{T-1} W'_{T-1}{}^{\mathbf{T}} H_{w^*}\right]$$
$$\qquad + \eta^2 \mathbb{E}_{V_T}\left[V_T V_T{}^{\mathbf{T}}\right],$$

where the last equation is by $\mathbb{E}_{V_T}^{w_{T-1}}[V_T] = 0$.

Recall that $\mathbb{E}_{V_T}\left[V_T V_T{}^{\mathbf{T}}\right] = C_T$ and notice that $\mathbb{E}_{W'_T}\left[W'_T W'_T{}^{\mathbf{T}}\right] = \mathbb{E}_{W'_{T-1}}\left[W'_{T-1} W'_{T-1}{}^{\mathbf{T}}\right] = \Lambda_{w^*}$ when $T \to \infty$ (i.e. ergodicity), we have

$$\Lambda_{w^*} H_{w^*} + H_{w^*} \Lambda_{w^*} - \eta H_{w^*} \Lambda_{w^*} H_{w^*} = \eta C_T.$$

Furthermore, if $H_{w^*}$ and $\Lambda_{w^*}$ commute, namely $\Lambda_{w^*} H_{w^*} = H_{w^*} \Lambda_{w^*}$, we have

$$[H_{w^*}(2\mathrm{I}_d - \eta H_{w^*})]\Lambda_{w^*} = \eta C_T,$$

which will give use $\Lambda_{w^*} = \eta\left[H_{w^*}(2\mathrm{I}_d - \eta H_{w^*})\right]^{-1} C_T$.

This completes the proof. □

### E.2  THEOREM E.1: A GENERAL BOUND

The following bound can be easily proved by using Eq. (14).

**Theorem E.1.** *Under the same conditions in Lemma 2.1 and Lemma 4.1, then for any* $P_{W_T} = \mathcal{N}\left(\tilde{w}, \widetilde{\Lambda}\right)$, *where $\tilde{w}$ and $\widetilde{\Lambda}$ are independent of S, we have*

$$\mathcal{E}_\mu(\mathcal{A}) \le \sqrt{\frac{R^2}{2n} \inf_{\tilde{w},\widetilde{\Lambda}} \mathbb{E}_{S,W_S^*}\left[\log \frac{\det\left(\widetilde{\Lambda}\right)}{\det\left(\Lambda_{W_S^*}\right)} + tr\left\{\widetilde{\Lambda}^{-1}\Lambda_{W_S^*} - \mathrm{I}_d\right\} + \mathrm{d}_\mathrm{M}^2\left(W_S^*, \tilde{w}; \widetilde{\Lambda}\right)\right]},$$

*where* $\mathrm{d}_\mathrm{M}\left(x, y; \Sigma\right) \triangleq \sqrt{(x-y)^T \Sigma^{-1}(x-y)}$ *is the Mahalanobis distance.*

### E.3 PROOF OF THEOREM 4.1

*Proof.* Let $P_{W_T} = \mathcal{N}\left(w_\mu^*, \Lambda_{w_\mu^*}\right)$, then

$$\mathbb{E}_{S,W_S^*}\left[\log \frac{\det\left(\Lambda_{w_\mu^*}\right)}{\det\left(\Lambda_{W_S^*}\right)} + tr\left\{\Lambda_{w_\mu^*}^{-1}\Lambda_{W_S^*} - \mathrm{I}_d\right\} + \left(W_S^* - w_\mu^*\right)^T \Lambda_{w_\mu^*}^{-1}\left(W_S^* - w_\mu^*\right)\right]$$

$$= \mathbb{E}_{S,W_S^*}\left[\log \frac{\det\left(\Lambda_{w_\mu^*}\right)}{\det\left(\Lambda_{W_S^*}\right)} + tr\left\{\Lambda_{w_\mu^*}^{-1}\Lambda_{W_S^*} - \mathrm{I}_d\right\} + tr\left\{\Lambda_{w_\mu^*}^{-1}\left(W_S^* - w_\mu^*\right)\left(W_S^* - w_\mu^*\right)^T\right\}\right]$$

$$= \mathbb{E}_{S,W_S^*}\left[\log \frac{\det\left(\Lambda_{w_\mu^*}\right)}{\det\left(\Lambda_{W_S^*}\right)}\right] + tr\left\{\Lambda_{w_\mu^*}^{-1}\mathbb{E}_{S,W_S^*}\left[\Lambda_{W_S^*}\right] - \mathrm{I}_d + \Lambda_{w_\mu^*}^{-1}\mathbb{E}_{W_S^*}\left[\left(W_S^* - w_\mu^*\right)\left(W_S^* - w_\mu^*\right)^T\right]\right\}.$$

(27)

Denote $\widetilde{\Sigma}_\mu \triangleq \mathbb{E}_{S,W_S^*}\left[\left(W_S^* - w_\mu^*\right)\left(W_S^* - w_\mu^*\right)^T\right] = \mathbb{E}_{W_S^*}\left[W_S^* W_S^{*T}\right] - w_\mu^* w_\mu^{*T}$.

Notice that

$$\mathbb{E}_{S,W_S^*}\left[\Lambda_{W_S^*}\right] = \mathbb{E}_{S,W_S^*,W_T}\left[\left(W_T - W_S^*\right)\left(W_T - W_S^*\right)^T\right]$$

$$= \mathbb{E}_{W_T}\left[W_T W_T^T\right] - \mathbb{E}_{W_S^*}\left[W_S^* W_S^{*T}\right]$$

$$= \mathbb{E}_{W_T}\left[W_T W_T^T\right] - w_\mu^* w_\mu^{*T} - \left(\mathbb{E}_{W_S^*}\left[W_S^* W_S^{*T}\right] - w_\mu^* w_\mu^{*T}\right)$$

$$= \Lambda_{w_\mu^*} - \widetilde{\Sigma}_\mu.$$

Therefore,

$$tr\left\{\Lambda_{w_\mu^*}^{-1}\mathbb{E}_{S,W_S^*}\left[\Lambda_{W_S^*}\right] - \mathrm{I}_d + \Lambda_{w_\mu^*}^{-1}\mathbb{E}_{W_S^*}\left[\left(W_S^* - w_\mu^*\right)\left(W_S^* - w_\mu^*\right)^T\right]\right\}$$

$$= tr\left\{\Lambda_{w_\mu^*}^{-1}\mathbb{E}_{S,W_S^*}\left[\Lambda_{W_S^*}\right] - \Lambda_{w_\mu^*}^{-1}\Lambda_{w_\mu^*} + \Lambda_{w_\mu^*}^{-1}\mathbb{E}_{W_S^*}\left[\left(W_S^* - w_\mu^*\right)\left(W_S^* - w_\mu^*\right)^T\right]\right\}$$

$$= tr\left\{\Lambda_{w_\mu^*}^{-1}\left(\mathbb{E}_{S,W_S^*}\left[\Lambda_{W_S^*}\right] - \Lambda_{w_\mu^*} + \widetilde{\Sigma}_\mu\right)\right\}$$

$$= 0.$$

Plugging this into Eq. (27), we have

$$\mathbb{E}_{S,W_S^*}\left[\log \frac{\det\left(\Lambda_{w_\mu^*}\right)}{\det\left(\Lambda_{W_S^*}\right)} + tr\left\{\Lambda_{w_\mu^*}^{-1}\Lambda_{W_S^*} - \mathrm{I}_d\right\} + \left(W_S^* - w_\mu^*\right)^T \Lambda_{w_\mu^*}^{-1}\left(W_S^* - w_\mu^*\right)\right]$$

$$= \mathbb{E}_{S,W_S^*}\left[\log \frac{\det\left(\Lambda_{w_\mu^*}\right)}{\det\left(\Lambda_{W_S^*}\right)}\right] = \mathbb{E}_{S,W_S^*}\left[tr\left\{\log\left(\Lambda_{W_S^*}^{-1}\Lambda_{w_\mu^*}\right)\right\}\right].$$

Finally, applying Theorem E.1 will conclude the proof. $\qquad\square$

### E.4 PROOF OF COROLLARY 4.1

*Proof.* The proof is straightforward by plugging $\Lambda_{w^*} = \left[ H_{w^*} \left( \frac{2}{\eta} \mathrm{I}_d \right) \right]^{-1} C_T$ in Theorem 4.1. $\square$

### E.5 PROOF OF COROLLARY 4.2

*Proof.* By Lemma C.2, it's easy to obtain the following bound according to Theorem 4.1.

$$\mathcal{E}_\mu(\mathcal{A}) \le \sqrt{\frac{R^2 d}{2n} \log \left( \frac{\mathbb{E}\left[\mathrm{d}_\mathrm{M}^2\left(W_S^*, w_\mu^*; \mathbb{E}\left[\Lambda_{W_S^*}\right]\right)\right]}{d} + 1 \right) + \mathbb{E}\left[ tr\left\{ \log\left(\Lambda_{W_S^*}^{-1} \mathbb{E}\left[\Lambda_{W_S^*}\right]\right) \right\} \right]}.$$

Then, plugging $\Lambda_{W_S^*} = \frac{\eta}{2b}\mathrm{I}_d$ will conclude the proof. $\square$

### E.6 COROLLARY E.1: DISTANCE TO INITIALIZATION

**Corollary E.1.** *Under (i-iii) in Lemma 4.1, then $\mathcal{E}_\mu(\mathcal{A}) \le \sqrt{\frac{dR^2}{n} \log\left( \frac{2b}{\eta d} \mathbb{E}||W_S^* - W_0||^2 + 1 \right)}$.*

*Proof.* Notice that $I(W_T; S) \le \mathbb{E}_S \mathrm{D}_\mathrm{KL}(Q_{W_T|S}||P_{W_T})$ holds for any $\sigma > 0$, then for a given $\tilde{w}$, we have

$$
\begin{aligned}
I(W_T; S) &= \inf_{P_{W_T}} \mathbb{E}_S \left[ \mathrm{D}_\mathrm{KL}(Q_{W_T|S}||P_{W_T}) \right] \\
&\le \inf_\sigma \mathbb{E}_S \left[ \mathrm{D}_\mathrm{KL}(P_{W_S^* + \sqrt{\frac{\eta}{2b}}N, W_S^*|S}||P_{\tilde{w}+\sigma N}) \right] && (28) \\
&= \inf_\sigma \mathbb{E}_{S,W_S^*} \left[ \mathrm{D}_\mathrm{KL}(P_{W_S^* + \sqrt{\frac{\eta}{2b}}N, |S, W_S^*}||P_{\tilde{w}+\sigma N}) \right] \\
&= \inf_\sigma \frac{1}{2}\mathbb{E}_{S,W_S^*} \left[ \frac{1}{\sigma^2}(W_S^* - \tilde{w})^T (W_S^* - \tilde{w}) + \log \frac{\sigma^{2d}}{(\eta/2b)^d} + tr\{\frac{\eta}{2b\sigma^2}\mathrm{I}_d\} - d \right] \\
&= \frac{1}{2}\inf_\sigma \frac{1}{\sigma^2}\mathbb{E}_{S,W_S^*} \left[ ||W_S^* - \tilde{w}||^2 + \frac{\eta d}{2b} \right] + d\log\sigma^2 + d\log\frac{2b}{\eta} - d \\
&= \frac{1}{2}d\log\left( \frac{2b}{\eta d}\mathbb{E}_{S,W_S^*}\left[||W_S^* - \tilde{w}||^2\right] + 1 \right), && (29)
\end{aligned}
$$

where Eq. (28) is by the chain rule of KL divergence, and the optimal $\sigma^* = \sqrt{\mathbb{E}_{S,W_S^*}\left[||W_S^* - \tilde{w}||^2/d + \frac{\eta}{2b}\right]}$. Let $\tilde{w} = W_0$ will conclude the proof. $\square$

Additionally, Corollary E.1 can be used to recover a trajectory-based bound.

**Corollary E.2.** *Let $W_T = W_s^*$, $\tilde{w} = 0$ and W.L.O.G, assume $W_0 = 0$, then*

$$\mathcal{E}_\mu(\mathcal{A}) \le \sqrt{\frac{dR^2}{n} \log\left( \frac{4bT\eta}{d} \sum_{t=1}^T \mathbb{E}\left[||G_t||^2 + tr\{C_t\}\right] + 1 \right)},$$

**Remark E.1.** *In Theorem 3.1, let $\tilde{g} = 0$ and by applying Jensen's inequality, we could also let the summation and factor $T$ move inside the square root. Then the most different part in Corollary E.2 is that $A_2(t)$ is now removed from the bound.*

*Proof.* When $W_0 = 0$, we notice that

$$W_T = \sum_{t=1}^T -\eta G_t + \eta N_{C_t},$$

where $N_{C_t} = C_t^{1/2} N_t$.

Thus,

$$||W_T||^2 = ||\sum_{t=1}^{T} -\eta G_t + \eta N_{C_t}||^2 \leq 2T\eta^2 \sum_{t=1}^{T} ||G_t||^2 + ||N_{C_t}||^2$$

Let $\tilde{w} = 0$, recall the bound in Corollary E.1 and plugging the inequality above, we have

$$
\begin{aligned}
\mathcal{E}_\mu(\mathcal{A}) &\leq \sqrt{\frac{R^2}{n} d \log\left(\frac{2b}{\eta d} \mathbb{E}_{S,W_T}\left[||W_T - \tilde{w}||^2\right] + 1\right)} \\
&\leq \sqrt{\frac{dR^2}{n} \log\left(4bT\eta/d\mathbb{E}_{S,W_{0:T-1},N_{C_{0:t-1}}}\left[\sum_{t=1}^{T}||G_t||^2 + ||N_{C_t}||^2\right] + 1\right)} \\
&= \sqrt{\frac{dR^2}{n} \log\left(\frac{4bT\eta}{d}\sum_{t=1}^{T}\mathbb{E}_{S,W_{t-1}}\left[||G_t||^2 + tr\{C_t\}\right] + 1\right)}
\end{aligned}
$$

This concludes the proof. $\square$

### E.7 PROOF OF THEOREM 4.2

*Proof.* Let $P_{W_T|S_J=s_j} = \mathcal{N}(W^*_{s_j}, \frac{\eta}{2b}I_d)$, then

$$
\begin{aligned}
D_{KL}(Q_{W_T|S=s}||P_{W_T|S_J=s_j}) &= D_{KL}(Q_{W^*_s+\sqrt{\frac{\eta}{2b}}N|S=s}||P_{W^*_{s_j}+\sqrt{\frac{\eta}{2b}}N|S_J=s_j}) \\
&\leq D_{KL}(Q_{W^*_s+\sqrt{\frac{\eta}{2b}}N,W^*_s|S=s}||P_{W^*_{s_j}+\sqrt{\frac{\eta}{2b}}N,W^*_{s_j}|S_J=s_j}) \quad (30) \\
&= \mathbb{E}_{W^*_s,W^*_{s_j}}\left[D_{KL}(Q_{W^*_s+\sqrt{\frac{\eta}{2b}}N|W^*_s,S=s}||P_{W^*_{s_j}+\sqrt{\frac{\eta}{2b}}N|W^*_{s_j},S_J=s_j})\right] \\
&= \mathbb{E}_{W^*_s,W^*_{s_j}}\left[\frac{b}{\eta}||W^*_s - W^*_{s_j}||^2\right], \quad (31)
\end{aligned}
$$

where Eq. (30) is by the chain rule of KL divergence. Plugging the Eq. (31) into Lemma 2.2 will obtain the final result. $\square$

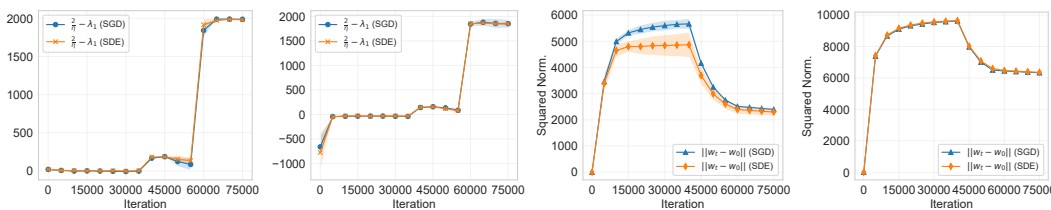

(a) ResNet on CIFAR10    (b) ResNet on CIFAR100    (c) ResNet on CIFAR10    (d) ResNet on CIFAR100

Figure 6: (a-b) The dynamics of $\eta/2 - \lambda_1$. Note that learning rate decays by 0.1 at the $40,000^{th}$ and the $60,000^{th}$ iteration. (c-d) The distance of current model parameters from its initialization.

## F EXPERIMENT DETAILS AND ADDITIONAL RESULTS

The implementation in this paper is on PyTorch (Paszke et al., 2019), and all the experiments are carried out on NVIDIA Tesla V100 GPUs (32 GB). Most experiment settings follow Wu et al. (2020), and the code is also based their implementation, which is available at: https://github.com/uuujf/MultiNoise.

### F.1 HYPERPARAMETERS

For CIFAR 10, the initial learning rates used for VGG-11 and ResNet-18 are $0.01$ and $0.1$, respectively. For SVHN, the initial learning rate is $0.05$. For CIFAR100, the initial learning rate is $0.1$. The learning rate is then decayed by $0.1$ at iteration $40,000$ and $60,000$. If not stated otherwise, the batch size of SGD is $100$.

### F.2 ADDITIONAL EMPIRICAL RESULTS

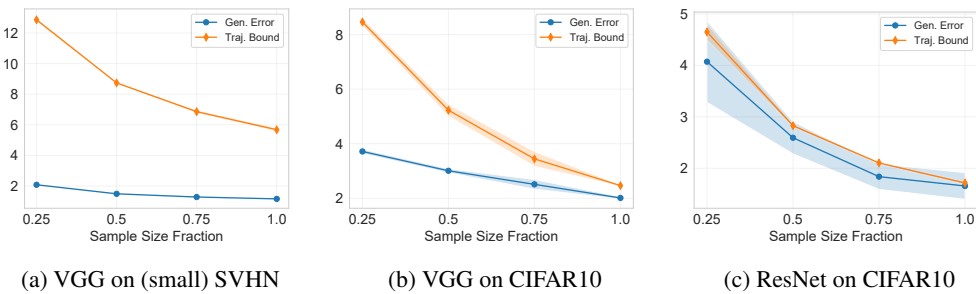

(a) VGG on (small) SVHN     (b) VGG on CIFAR10     (c) ResNet on CIFAR10

Figure 7: Zoomed-in of generalization error.

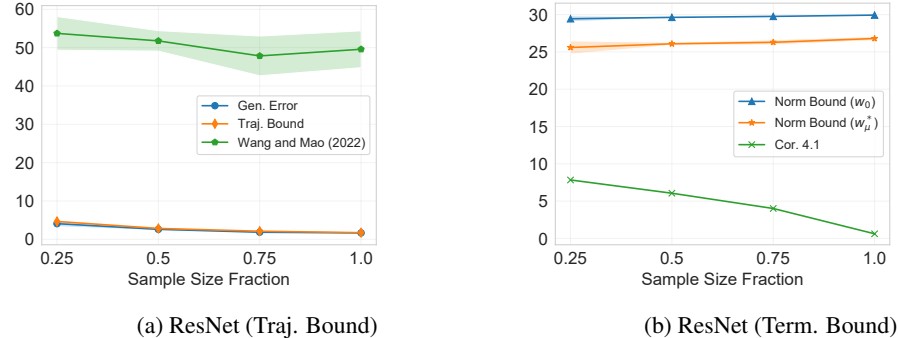

(a) ResNet (Traj. Bound)        (b) ResNet (Term. Bound)

Figure 8: Estimated trajectories-based bound and terminal-state based bound, with $R$ excluded. Models trained on CIFAR 10.

## G ADDITIONAL RESULT: INVERSE POPULATION FIM AS BOTH POSTERIOR AND PRIOR COVARIANCE

Inspired by some previous works of (Achille et al., 2019; Harutyunyan et al., 2021; Wang et al., 2022), we can also select the inverse population Fisher information matrix $F_{w^*}^\mu = \mathbb{E}_Z \left[ \nabla \ell(w^*, Z) \nabla \ell(w^*, Z)^T \right]$ as the posterior covariance. Then, the following theorem is obtained.

**Theorem G.1.** *Under the same conditions in Theorem 4.2, and assume the distribution $P_{W_{\hat{S}_J}|S_J}$ is invariant of $J$, then*

$$\mathcal{E}_\mu(\mathcal{A}) \le \frac{M}{2n} \mathbb{E}_S \left[ \sqrt{\mathbb{E}_{W_{\hat{S}}^*}^S \left[ tr\{ H_{W_{\hat{S}}^*}^{-1} F_{W_{\hat{S}}^*}^\mu \} \right]} \right].$$

**Remark G.1.** *Notice that $F_{W_{\hat{S}}^*}^\mu \approx H_{W_{\hat{S}}^*}^\mu \approx \Sigma^\mu(W_{\hat{S}}^*)$ near minima (Pawitan, 2001, Chapter 8), then $tr\{ H_{W_{\hat{S}}^*}^{-1} \Sigma^\mu(W_{\hat{S}}^*) \}$ is very close to the Takeuchi Information Criterion (Takeuchi, 1976). In addition, our bound in Theorem G.1 is similar to Singh et al. (2022, Theorem 3.) with the same convergence rate, although strictly speaking, their result is not a generalization bound. Moreover, as also pointed out in Singh et al. (2022), here $H_{W_{\hat{S}}^*}^{-1}$ is evaluated on the training sample, unlike other works that evaluates the inverse Hessian on the testing sample (e.g., Thomas et al. (2020)).*

The invariance assumption is also used in Wang et al. (2021a). In practice, $n$ is usually very large, when $m = n - 1$, this assumption indicates that replacing one instance in $s_j$ will not make $P_{W_{s_j}^*|s_j}$ be too different.

## H    PROOF OF THEOREM G.1

*Proof.* We now use $(F_{W_S^*}^\mu)^{-1}$ as both the posterior and prior covariance (again, we assume $F_{W_S^*}^\mu \approx F_{W_{S_j}^*}^\mu$ for any $j$), then

$$
\begin{aligned}
\mathcal{E}_\mu(\mathcal{A}) \leq & \mathbb{E}_S\left[\sqrt{\frac{M^2}{4}\mathbb{E}_{J,W_S^*,W_{S_J}^*}^S\left[\left(W_S^* - W_{S_J}^*\right)F_{W_S^*}^\mu\left(W_S^* - W_{S_J}^*\right)^T\right]}\right] \\
= & \frac{M}{2n}\mathbb{E}_S\left[\sqrt{\mathbb{E}_{W_S^*,W_{S_j}^*}^S\left[tr\left\{F_{W_S^*}^\mu H_{W_S^*}^{-1} H_{W_S^*}^{-1}\mathbb{E}_J\left[\nabla\ell(W_S^*,Z_i)\nabla\ell(W_S^*,Z_i)^T\right]\right\}\right]}\right] \\
= & \frac{M}{2n}\mathbb{E}_S\left[\sqrt{\mathbb{E}_{W_S^*}^S\left[tr\left\{F_{W_S^*}^\mu H_{W_S^*}^{-1}\right\}\right]}\right],
\end{aligned}
$$

which completes the proof. $\qquad\square$

