# OpenReview forum: "Two Facets of SDE Under an Information-Theoretic Lens: Generalization of SGD via Training Trajectories and via Terminal States"
_ICLR.cc/2024/Conference — Submitted to ICLR 2024_

### Official Review · Reviewer_4Y6d · 2023-11-01

**Soundness:** 3 good
**Presentation:** 3 good
**Contribution:** 3 good
**Rating:** 6
**Confidence:** 4

**Summary:**

- The authors provided an information-theoretic generalization error bound for SGD by utilizing the interesting connection between SGD and SDE discussed in [Mandt et al. (2017)] and [Jastrzebski et al. (2017)], etc..
- They showed the close relationship between the population gradient covariance and the covariance of the gradient noise, thus justifying the significance of the trace of gradient noise covariance in the generalization ability of SGD.
- Additionally, we applied the obtained information-theoretic bounds to derive the generalization error upper bound for SGD based on distribution-dependent prior distributions or data-dependent prior distributions. Using these results, they derived bounds based on alignment between the weight covariance matrix for each individual local minimum and the weight covariance matrix for the average of local minima, as well as bounds based on sensitivity.
- They empirically confirmed the generalization ability of SGD/SGE and the key components within the derived bounds for both algorithms by utilizing their generalization bounds, revealing that these components for SGD and SDE align remarkably well. They also showed that their bound is tigther than that of [Wang & Mao (2022)].

**Strengths:**

- Information-theoretic generalization error analysis is particularly effective for analyzing noisy and iterative algorithms like SGLD. However, in the case of SGD, the upper bounds for mutual information (MI) diverge, making it challenging to apply directly. In this paper, instead of relying on the commonly used auxiliary process technique, the authors enabled information-theoretic generalization error analysis by exploiting the connection between SGD and SDE using full-batch gradients and mini-batch gradients, as a means to address this issue. This is a very intriguing approach with broad potential applications.
- Through the derived generalization error bounds, this paper provided a theoretical validity for empirically known discussions of the impact of gradient variance on generalization performance and the improvement of generalization performance through control of gradient norms.
- In the experiments, the analysis not only focuses on the tightness of the bounds but also examines the numerical assessable elements that constitute these bounds. Furthermore, it confirms the close agreement between SGD and SDE. These factors collectively ensure the validity of the theoretical results.
- From the above, it is evident that this paper is well-written and provides a significant impact in the research field of generalization performance analysis obtained through stochastic optimization.

**Weaknesses:**

- The limitations of the information-theoretic generalization error analysis approach, such as (implicit/explicit) dimension-dependence and time-dependence, are inherent in the bounds of this paper (although mitigating these limitations can be challenging or important future work). The paper would have been better if there had been a part discussing the limitations and future prospects related to these issues.
- The approximation of $\Lambda_{w*}$ leads to a dependency on the inverse of the learning rate $\eta$ in the bounds provided in Corollaries 4.1 and 4.2, as well as Theorem 4.2. It seems that these bounds become large or even diverge as the learning rate decreases ($\eta_t \rightarrow 0$). A similar problem appears in the sensitivity-based generalization error analysis for SGLD [1]. I guess that the reason for decaying the learning rate only between the 40,000th and 60,000th iterations in the experiments is to prevent the bounds from diverging, as reducing the learning rate too early might cause divergence before the parameters have converged sufficiently.

**Questions:**

I would like to express my sincere respect for all the efforts the authors have invested in this paper.
In connection with the weaknesses mentioned above, I would like to pose several questions related to the concerns raised.
I would appreciate your responses.

- In this paper, the authors make the assumption of a loss function that is both differentiable and satisfies the $R$-subGaussian property. Can you provide specific examples of loss functions that meet both of these assumptions, excluding bounded losses?
- As I comprehend it, the generalization error bounds presented in Theorem 3.1, Corollary 3.1, Theorem 3.2, Theorem 4.1, and so forth, within this paper, explicitly or implicitly rely on the parameter dimension $d$. I'm curious about the behavior of these bounds as $d$ grows ($d \rightarrow \infty$). Especially, in cases where dimensionality dependence is observed both inside and outside the logarithmic expressions, it appears that these bounds tend to diverge unless factors such as gradient noise are adequately small. Could you please share the authors' thoughts on this matter?
- Is it feasible to find an approach for approximating $\Lambda_{w*}$ that eliminates the reliance on the inverse of $\eta$?
Is delaying the timing of reducing the learning rate the sole method when numerically evaluating your bounds? I believe there is some relationship between the generalization (or the desired convergence) and the speed of learning rate decay. What is your perspective on these concerns?

## MISC
- When numerically evaluating generalization errors in experiments, I imagine it involves evaluating the difference in accuracy between the test data and the training data. In such cases, even if the training/test accuracy is low, it's possible for the apparent generalization error to be small. Therefore, if possible, reporting the predictive accuracy after training completion would enhance the reliability of the experimental results (I refer to Appendices H and I in [2] for example).
- p.4, the paragraph of "Validation of SDE," the 2nd paragraph: an oder 1 strong... --> an order 1 strong... ?

## Citation
(Note: I am not the author of the following papers)

[1]: T. Farghly and P. Rebeschini. Time-independent Generalization Bounds for SGLD in Non-convex Settings. In NeurIPS2021.

[2](The authors have already cited): J. Negrea, M. Haghifam, G. K. Dziugaite, A. Khisti, and D. M Roy Information-theoretic generalization bounds for SGLD via data-dependent estimates. In Advances in Neural Information Processing Systems, 2019.

**Details Of Ethics Concerns:**

I believe that this work does not raise any ethical concerns because it is a theoretical study focused on the generalization ability obtained by SGD.

---

> ### Author Response · Authors · 2023-11-18
> **To Reviewer 4Y6d**
>
> We thank you sincerely for your comments to our paper. Our responses follow.
>
> >- The limitations of the information-theoretic generalization error analysis approach, such as (implicit/explicit) dimension-dependence and time-dependence ...
>
> **Response.** Thank you for this valuable suggestion. We have included a section on limitations and future works to discuss the dimension-dependence. Additionally, we provided additional elaboration on the time-dependence towards the end of Section 3, and consider the time-dependence as motivation for introducing terminal-state bounds in Section 4, which, by their asymptotic nature, are time-independent.
>
>
> >- The approximation of $\Lambda_{w^*}$ leads to a dependency on the inverse of the learning rate $\eta$ in the bounds provided in Corollaries 4.1 and 4.2, as well as Theorem 4.2 ...
>
> **Response.**  We would like to clarify that Corollaries 4.1-4.2 and Theorem 4.2 present terminal-state-based bounds. The learning rate considered in these bounds corresponds to the one applied in the final iteration. Furthermore, we think that the reverse relationship between the generalization bound and the learning rate is more favorable. Such bounds suggest that a larger learning rate may lead to better generalization, which aligns with various empirical observations.
>
>  >- In this paper, the authors make the assumption of a loss function that is both differentiable and satisfies the $R$-subGaussian property. Can you provide specific examples of loss functions that meet both of these assumptions, excluding bounded losses?
>
> **Response.** At present, we are not aware of a concrete example of this kind of loss beyond the bounded case. To our knowledge, in order to satisfy the sub-Gaussian assumption, a typical choice in the literature is a bounded cross-entropy loss, such as $-\ln(e^{-\ell_{\max}}+(1-2e^{-\ell_{\max}}))f(x)[y]$, where $f(x)[y]$ is the probability assigned to the label $y$ by $f$ for $x$, as given in [3].
>
> Furthermore, despite the unbounded nature of the original cross-entropy loss, we believe that it indeed has a subGaussian distribution under the SGD training. Specifically, by using the prevalent training methodologies, one can typically ensure that the loss curve consistently descends until it reaches convergence, effectively bounding the loss values during training by the initial value. We have not made an effort in rigorously proving the subGaussianality of the loss. Such an effort would require a careful examination of the SGD dynamics and involve significant technicality.
>
> [3] Dziugaite and Roy. "Data-dependent PAC-Bayes priors via differential privacy." NeurIPS 2018.
>
> >- As I comprehend it, the generalization error bounds presented in Theorem 3.1, Corollary 3.1, Theorem 3.2, Theorem 4.1, and so forth, within this paper, explicitly or implicitly rely on the parameter dimension $d$ ...?
>
> **Response.** We note that expressions such as $d\log{\frac{Y}{d}}$ may have an $d$-independent upper bound by the inequality $\log(x+1)\leq x$. For example, in Corollary 3.1 of our paper, as long as the gradient norm is upper bounded by a constant, Corollary 3.1 will vanish regardless of the choice of $d$.
>
> However, we acknowledge that most of our bounds are inherently dimension-dependent, so there exist some settings of $d$ and $n$, particularly when $d$ grows faster than $n$ (e.g., $d>\mathcal{O}(n^2)$), such that these bounds becomes vacuous. Such limitations exist in nearly all the information-theoretic generalizations and sometimes lead these bounds to non-vanishing in the stochastic convex optimization, as recently explored in [4]. While these limitations are not easily overcome fundamentally, the nature of non-convexity in deep learning may provide alternative potentials. For example, although deep neural networks are often overparameterized, recent research, as highlighted in works such as [5,6,7], reveals that SGD/GD only operates within a subspace of the neural networks, referred to as the *intrinsic dimension*. In this context, the SDE in Eq.(5) or $C_t$ itself can be defined within these invertible subspaces, and our derivations remain valid. Moreover, the theoretical characterization of the *intrinsic dimension* remains an open problem. The further development of this research direction will undoubtedly enhance and contribute to the advancements in our work.
>
> [4] Haghifam et al. "Limitations of Information-Theoretic Generalization Bounds for Gradient Descent Methods in Stochastic Convex Optimization." ALT 2023.
>
> [5] Li et al. "Measuring the Intrinsic Dimension of Objective Landscapes." ICLR 2018.
>
> [6] Gur-Ari et al. "Gradient descent happens in a tiny subspace." arXiv preprint arXiv:1812.04754 (2018).
>
> [7] Larsen et al. "How many degrees of freedom do we need to train deep networks: a loss landscape perspective." ICLR 2022.

---

> > ### Author Response · Authors · 2023-11-18
> > **To Reviewer 4Y6d (cont.)**
> >
> > >- Is it feasible to find an approach for approximating $\Lambda_{w^*}$ that eliminates the reliance on the inverse of $\eta$? Is delaying the timing of reducing the learning rate the sole method when numerically evaluating your bounds? I believe there is some relationship between the generalization (or the desired convergence) and the speed of learning rate decay. What is your perspective on these concerns?
> >
> > **Response.** As mentioned in the previous response, in our opinion, we think that generalization bounds that reverse-depend on the learning rate are more consistent with empirical observations in practice. Additionally, it's worth noting that the learning rate decay strategy does not affect the numerical evaluation of the bounds in Section 4 since they use the final learning rate.
> >
> > While we agree with the reviewer that the speed of learning rate decay can impact the optimization and generalization of SGD, our terminal-state bounds might be insufficient to provide meaningful insights into this matter at this point. However, it is essential to highlight that our trajectory-based bounds, capable of tracking the training dynamics, hold potential for investigating the effects of the learning rate. Despite not explicitly incorporating the learning rate, studying how learning rate decay affects the structure of gradient noise might yield valuable insights.
> >
> > >- When numerically evaluating generalization errors in experiments, I imagine it involves evaluating the difference in accuracy between the test data and the training data. In such cases, even if the training/test accuracy is low, it's possible for the apparent generalization error to be small. Therefore, if possible, reporting the predictive accuracy after training completion would enhance the reliability of the experimental results (I refer to Appendices H and I in [2] for example).
> >
> > **Response.** Thank you for this suggestion and thanks for bringing these experiments to our attention, we will try to include them in the next revision.
> >
> > >- p.4, the paragraph of "Validation of SDE," the 2nd paragraph: an oder 1 strong... --> an order 1 strong... ?
> >
> > **Response.** Thanks for your careful reading, we have fixed this.

---

> ### Comment · Reviewer_4Y6d · 2023-11-23
> **Acknowledgement**
>
> Thank you for the constructive discussion, corrections, and responses to my concerns.
> Although the presentation of the terminal state results remains an issue as Reviewer DrRU said, I remain of the opinion that this paper is an important result in the area of IT-based analysis in SGD.
> Therefore, I will keep my score. I wish the authors the best of luck.
>
> Sincerely,
>
> Reviewer 4Y6d

---

### Official Review · Reviewer_2gqG · 2023-11-01

**Soundness:** 3 good
**Presentation:** 3 good
**Contribution:** 3 good
**Rating:** 6
**Confidence:** 2

**Summary:**

This work proposes several new information-theoretic generalization bounds based on an SDE approximation of SGD. Detailed discussions on these results are provided, including a comparison with prior information-theoretic bounds and specific corollaries for different scenarios (e.g., isotropic vs. anisotropic prior). All the results are validated experimentally throughout.

**Strengths:**

1. Extensive discussions on prior works and the context behind considering information-theoretic bounds + thorough literature survey
2. Experimental verifications support the theoretical bounds

**Weaknesses:**

1. The overall organization can be done better. For instance, there are many different information-theoretic bounds throughout the papers, each separately discussed below the corresponding theorems. It would be very helpful for first-time readers if the paper had a summarizing section that gathers and summarizes all the results and compares them (e.g. when is this more useful/tighter)
2. Overall, although the results are better than previous bounds (e.g., Neu et al. (2021)), to me, it's a bit unclear exactly what are the technical novelties in allowing for better results. The discussions provided after each theorem helped me understand the context of how to interpret the new result, but I'm a bit confused about whether the results themselves are new or are just improvements of known bounds. Even in the contributions, although the paper provides much explanation on what the bounds mean and what they are saying, the novelty of the results is somewhat unclear (As most of the implications are already somewhat known in deep learning theory literature).

**Questions:**

1. Overall, am I correct in saying that the paper proposes analyses that provide a somewhat unifying information-theoretic perspective on the folklore (some of which have been studied extensively) results in deep learning theory?
2. Can the authors provide the plot of the proposed generalization bounds by somehow estimating the mutual information?
3. Can this be extended to other optimizers such as momentum, adam, adagrad...etc?
4. Any connection to the information bottleneck theory of deep learning?
5. Can the authors do something similar with noisy SGD [1]?
6. Corollary C.1 states that the generalization is controlled by the distance from initialization. Does this mean that lazy training is when generalization is the best? Or is it that the lazy training phase is not compatible with the required assumptions?


[1] https://proceedings.mlr.press/v178/vivien22a/vivien22a.pdf

---

> ### Author Response · Authors · 2023-11-18
> **To Reviewer 2gqG**
>
> We thank you sincerely for your valuable comments to our paper. Our responses follow.
>
> >- The overall organization can be done better. For instance, ....
>
> **Response.** Thank you for this suggestion. We have included Table 1 in the Appendix of the revision, which summarizes all the obtained bounds in this paper. Additionally, we have provided important remarks about these bounds in the table.
>
> >- Overall, although the results are better than previous bounds (e.g., Neu et al. (2021)), to me, it's a bit unclear ...
>
> **Response.** We would like to illustrate the reason that our bounds are better than Neu et al. (2021). Specifically, they invoke an auxiliary weight process, we let $\mathcal{A}\_{AWP}$ to denote the auxiliary noisy weight process in Neu et al. (2021). In our paper, we use SDE in Eq.(5), denoted as $\mathcal{A}\_{SDE}$. Let $\mathcal{A}\_{SGD}$ denote the original algorithm of SGD, recall $\mathcal{E}$ is the generalization error, then Neu et al. gives the bound as
> $$
> \mathcal{E}(\mathcal{A}\_{SGD})=\mathcal{E}(\mathcal{A}\_{SGD})+\mathcal{E}(\mathcal{A}\_{AWP})-\mathcal{E}(\mathcal{A}\_{AWP})\leq \text{GenBound}(\mathcal{A}\_{AWP})+|\mathcal{E}(\mathcal{A}\_{SGD})-\mathcal{E}(\mathcal{A}\_{AWP})|.
> $$
>
> Our paper gives the bound
> $$
> \mathcal{E}(\mathcal{A}\_{SGD})=\mathcal{E}(\mathcal{A}\_{SGD})+\mathcal{E}(\mathcal{A}\_{SDE})-\mathcal{E}(\mathcal{A}\_{SDE})\leq \text{GenBound}(\mathcal{A}\_{SDE})+|\mathcal{E}(\mathcal{A}\_{SGD})-\mathcal{E}(\mathcal{A}\_{SDE})|.
> $$
>
> Empirical evidence, as shown in Figure 1 in our paper and in previous works such as [2,3], indicates that  $|\mathcal{E}(\mathcal{A}\_{SGD})-\mathcal{E}(\mathcal{A}\_{SDE})|$ is nearly negligible, particularly when compared to the scale of the bound in our result. On the other hand, it remains uncertain that whether the residual term $|\mathcal{E}(\mathcal{A}\_{SGD})-\mathcal{E}(\mathcal{A}\_{AWP})|$ for an artificial $\mathcal{A}\_{AWP}$ in Neu et al. (2021)  is small. This is a crucial factor contributing to the improvement in our bound over theirs. To be fair,  $\mathcal{A}\_{AWP}$ can be optimized over the Gaussian variance, but this might be challenging beyond the isotropic covariance case studied in Wang and Mao (2022).
>
> In terms of technical novelty, while the developments in $\text{GenBound}(\mathcal{A}\_{SDE})$ and $\text{GenBound}(\mathcal{A}\_{AWP})$ are closely related, both rooted in [4], previous works, including Neu et al. (2021), analyze both data-independent and state-independent isotropic noise covariance. In contrast, our SDE uses a data-dependent and state-dependent anisotropic noise covariance, rendering some crucial steps not directly applicable. Due to the different noise structure, the step of selecting a "good" prior distribution differs significantly. Additionally, we apply a variant of the log-sum inequality to demonstrate that the population gradient noise covariance is superior to the isotropic covariance, a method not invoked before. Furthermore, prior information-theoretic bounds have not considered the posterior distribution as a mixture of stationary Gaussians, as presented in our Section 4.
>
> [2] Wu et al. "On the noisy gradient descent that generalizes as sgd." ICML 2020.
>
> [3] Li et al. "On the validity of modeling sgd with stochastic differential equations (sdes)." NeurIPS 2021.
>
> [4] Pensia et al. "Generalization error bounds for noisy, iterative algorithms." ISIT 2018.
>
> >- Overall, am I correct in saying that the paper proposes analyses that provide a somewhat unifying ...?
>
> **Response.** While our findings do align or link with certain prior results in the broader landscape of deep learning theory, it's essential to emphasize that our paper introduces novel aspects. For instance, our Theorem 3.2, demonstrating that the alignment between population gradient noise covariance and batch gradient noise covariance can serve as an indicator for generalization, has not been explored before.
>
> >- Can the authors provide the plot of ...?
>
> **Response.** This is a good suggestion to verify the numerical tightness between the bounds and $I(W;S)$. However, it may not be feasible to estimate $I(W;S)$ in the most relevant cases, given the challenges associated with their high dimensionality and the continuous nature of these random variables. It's worth noting that even with advanced mutual information estimators like MINE, uncertainties about the approximation error could render the comparison potentially meaningless.

---

> > ### Author Response · Authors · 2023-11-18
> > **To Reviewer 2gqG (cont.)**
> >
> > >- Can this be extended to other optimizers such as momentum, adam, adagrad...etc?
> >
> > **Response.** It is possible to extend to the momentum case but may require some necessary simplifications as used in [4] and Neu et al. (2021). Extending to Adam would likely require a substantially different SDE approximation, which might not be straightforward. Fortunately, the SDE of Adam has been recently explored in [5], providing a valuable starting point. We have acknowledged these potential extensions in the future work section of our revised paper.
> >
> > [5] Malladi et al. "On the SDEs and scaling rules for adaptive gradient algorithms." NeurIPS 2022.
> >
> > >- Any connection to the information bottleneck theory of deep learning?
> >
> > **Response.** We believe that the notion of establishing such a connection is likely on the minds of most researchers studying information-theoretic bounds, and it still remains a challenge. Bridging this gap could involve decomposing the original $I(W;Z)$ in information-theoretic bounds into $I(W;X)$ and $I(W;Y|X)$ using the chain rule. While IB theory proposes that increasing $I(W;X)$ is beneficial for reducing training error and controlling $I(W;Y|X)$ is crucial to prevent overfitting, the precise connection with the SGD or SDE process remains uncertain to us.
> >
> > >- Can the authors do something similar with noisy SGD [1]?
> >
> > **Response.** Thank you for bringing this paper into our attention. Although label noise gradient descent is introduced in Section 2.2, their primary focus is on investigating the continuous model, specifically the label noise stochastic gradient flow. Given this, the analytic methods used in [6] might be more suitable. Additionally, our initial exploration of the discrete model presented in their Section 2.2 has proven challenging to interpret, yielding no useful insights thus far. We will continue to explore this aspect.
> >
> > [6] Mou et al. "Generalization bounds of sgld for non-convex learning: Two theoretical viewpoints." COLT 2018.
> >
> > >- Corollary C.1 states that the generalization is controlled by the distance from initialization. Does this mean that lazy training is when generalization is the best? Or is it that the lazy training phase is not compatible with the required assumptions?
> >
> > **Response.** Under the precondition that overparameterized models can attain zero training loss, we think our results do indicate that lazy training can lead to good generalization.

---

> > > ### Comment · Reviewer_2gqG · 2023-11-23
> > >
> > > I apologize for not being very active in the discussion, and I would like to thank the authors for answering my concerns.
> > >
> > > After reading the other reviews and reading the response to my review, I am satisfied with the authors' response, the paper's novelty, and the reorganization (among other modifications that the authors have made throughout the discussion period). In light of these points, I'm raising my score to 6.

---

### Official Review · Reviewer_DrRU · 2023-11-01

**Soundness:** 3 good
**Presentation:** 3 good
**Contribution:** 2 fair
**Rating:** 5
**Confidence:** 4

**Summary:**

This paper explores the challenge of establishing generalization bounds for SGD. It does so by using information-theoretic measures to assess the connection between a learning algorithm's output and its input. These bounds rely on two key assumptions:

1- Approximating SGD using a discrete Stochastic Differential Equation (SDE) framework, where the "minibatch noise" is Gaussian with a data-dependent covariance matrix.

2- Assuming that the transition kernel of SGD follows a Gaussian distribution.

Using these assumptions, the authors provide generalization bounds by characterizing the mutual information between the learning algorithm's output and its training dataset.

**Strengths:**

I think providing generalization bound for SGD is an important problem. Approximating SGD update rule with anisotropic SGLD seem an interesting idea and the bounds are intuitive.

**Weaknesses:**

Motivation of this work is not clear for me. It seems the main motivation is the connections between dynamics of SGD and its associated discrete SDE. The only evidence in the paper is a plot in the introduction. In the other related work, the authors miss many prior work and also the discussion in the related work section seems in-complete. Validation of SDE section in the paper also include lots of technical terms without exactly defining them.

Regarding Section 3 of the paper, many parts are not clear to me. In particular, it is assumed that the transition kernel of SGD is Gaussian around the minima. why is it a valid assumption? Transition kernel of SGD is just a bunch of delta measures based on the different realization of the mini-batches.

In general, I found the motivation of the paper is rather weak.

**Questions:**

- What does it mean that the distribution around local minima be Gaussian distribution? I do not understand the assumptions in Section 3.

- The statement of the theorems can be improved. For instance, it is difficult to understand the assumptions of Theorem 4.1.

- An important related work is

Wang, Bohan, et al. "Optimizing information-theoretical generalization bound via anisotropic noise of SGLD." Advances in Neural Information Processing Systems 34 (2021): 26080-26090.

In this paper the authors also consider the problem of obtaining generalization bounds for SGD+ anisotropic noise. What are the technical differences between the results in this paper and Wang et al?

---

> ### Author Response · Authors · 2023-11-18
> **To Reviewer DrRU**
>
> We thank you sincerely for your valuable feedback on our paper. Our responses follow.
>
> >- Motivation of this work is not clear for me ....
>
> **Response.** We would like to clarify the motivation behind our work. Analyzing SGD directly through information-theoretic bounds is often intractable due to its discrete nature (i.e. leading to a degenerate distribution in KL divergence), as also mentioned in your comments. However, the SDE approximation, where the updating process is smoothed by Gaussian noise, is amenable to analysis through information-theoretic bounds. Specifically, let $\mathcal{A}\_{SGD}$ denote the original algorithm of SGD, and let $\mathcal{A}\_{SDE}$ be the SDE in Eq.(5),  recall $\mathcal{E}$ is the generalization error, then $\mathcal{E}(\mathcal{A}\_{SGD})=\mathcal{E}(\mathcal{A}\_{SGD})+\mathcal{E}(\mathcal{A}\_{SDE})-\mathcal{E}(\mathcal{A}\_{SDE})\leq \text{GenBound}(\mathcal{A}\_{SDE})+|\mathcal{E}(\mathcal{A}\_{SGD})-\mathcal{E}(\mathcal{A}\_{SDE})|$.
>
> The key observation that $|\mathcal{E}(\mathcal{A}{SGD})-\mathcal{E}(\mathcal{A}{SDE})|$ is nearly negligible, as demonstrated in Figure 1, encourages our focus on analyzing the generalization bound of SDE. We also note that similar empirical evidence has been observed in some previous studies, such as [1,2].
>
> To provide further clarity on our motivation, we have added an additional section in Appendix B.1.
>
> [1] Wu et al. "On the noisy gradient descent that generalizes as sgd." ICML 2020.
>
> [2] Li et al. "On the validity of modeling sgd with stochastic differential equations (sdes)." NeurIPS 2021.
>
> >- In the other related work, the authors miss many prior work and also the discussion in the related work section seems in-complete. Validation of SDE section in the paper also include lots of technical terms without exactly defining them.
>
> **Response.** We have augmented the section on related works in our paper and expanded on the validation of SDE in Appendix B.2. We note that previous theoretical results aim to illustrate that SDE can weakly approximate SGD in the distribution sense, rather than precisely matching any single sample path of SGD. In addition, although we have made an effort to include all related works, there is a chance that we may have overlooked some important prior works. We genuinely welcome the reviewer to identify any such works, and we would be more than willing to incorporate them into our paper.
>
> >- Regarding Section 3 of the paper, many parts are not clear to me. ...
> >- What does it mean that the distribution around local minima be Gaussian distribution? I do not understand the assumptions in Section 3.
>
> **Response.** We think the reviewer might be referring to Section 4 rather than Section 3, as the Gaussian distribution assumption around the local minima is used in Section 4 (please correct us if we have misunderstood anything). We have provided this background information in Appendix B.3. In the continuous case, this is rooted in a classical result of the Ornstein–Uhlenbeck process, which is a Gaussian process. The stationary solution to its corresponding Fokker–Planck equation yields a Gaussian density. In the discrete case, in the long-time limit, $W_T$ represents the weighted sum of Gaussian noises. It's important to note that in both cases, we require the quadratic approximation of the loss (i.e., Eq. (7)).
>
> >- The statement of the theorems can be improved. For instance, ...
>
> **Response.** We have revised the statement to make it be more reader-friendly. Please do let us know if any further improvements are needed or if there are specific aspects that remain challenging to follow.

---

> ### Author Response · Authors · 2023-11-18
> **To Reviewer DrRU (cont.)**
>
> >- An important related work is
> Wang, Bohan, et al. "Optimizing information-theoretical generalization bound via anisotropic noise of SGLD." Advances in Neural Information Processing Systems 34 (2021): 26080-26090.
> In this paper the authors also consider the problem of obtaining generalization bounds for SGD+ anisotropic noise. What are the technical differences between the results in this paper and Wang et al?
>
> **Response.** We have included a discussion on the comparison with Wang et al. (2021) in Appendix D.7 (refer to the discussions after Lemma D.4). This work is indeed closely related, and it was an oversight on our part to merely mention it without elaborating on the differences.
>
> To clarify, their paper studies the algorithm of SGD with anisotropic noise, while our SDE analysis focuses on GD with anisotropic noise. This means that the discrete gradient noise arising from mini-batch sampling still exists in their analyzed algorithm, whereas the gradient noise is fully modeled as Gaussian in our Section 3. Moreover, Wang et al. (2021) uses matrix analysis tools to optimize the prior distribution. A significant distinction lies in their optimization analysis, which relies on the assumption that the trace of gradient noise covariance remains unchanged during training (see **Constriant 1** in their paper). Additionally, their final optimal posterior covariance is derived based on the assumption that the posterior distribution of $W$ is invariant to the data index, see Assumption 1 in their paper (to be fair, we also uses this assumption to give an additional result in Appendix, i.e. Theorem G.1). In contrast, our Section 3 avoids making these assumptions and demonstrates the superiority of population gradient noise covariance (GNC) in Lemma 3.2, by invoking a variant of the log-sum inequality. In summary, our proof is simpler and more straightforward, while Wang et al. (2021) makes a stronger claim about the optimality of population GNC based on their additional assumptions.

---

### Official Review · Reviewer_E5SG · 2023-11-02

**Soundness:** 2 fair
**Presentation:** 2 fair
**Contribution:** 2 fair
**Rating:** 6
**Confidence:** 3

**Summary:**

The paper analyzes noisy gradient descent where the added noise is an anisotropic, state dependent Gaussian noise. Two bounds on the expected generalization error of this algorithm are established using tools from information theory. The first is a trajectory dependent bound, the second takes a quadratic approximation of the loss around a local minimum and derives a result for noisy GD in that basin. In addition, some experiments are conducted to argue that the noisy gradient descent scheme constitutes a valid approximation of SGD.

**Strengths:**

The paper applies the idea of using data depended priors to bound mutual information to study a noisy gradient descent scheme that appears to not have been studied before.

**Weaknesses:**

- Some inappropriate comparisons and missing discussions: The bounds in this paper are established for noisy GD, but comparisons to [1] and [2], which is are results on SGD analyzed through an auxiliary sequence are made. It is strange to say a bound on a different object is tighter. More important are the missing discussions of other data dependent bounds established for schemes very similar to the studied noisy GD. For example, the very paper this work builds on has a result on SGLD where the batch noise covariance appears [3, theorem 3.1], comparisons with such results would be more appropriate. Especially since theorem 3.1 in this work is a straightforward application of the method in [3].
- Interpretability of the bounds: The choice of priors would ideally yield meaningful quantities. Here, aside from obtaining a dependence on trajectory wise gradient norms, a result already known for generic noisy iterative schemes, the paper proves Theorem 3.2 where the alignment between the batch estimate covariance and the population gradient covariance appears. It is difficult to understand why this quantity is interesting as it is not well discussed. Moreover, there should be more discussion on the positive definite assumption because, in practice, for neural networks, there are many more weights than data points, thus making $C_t$ invertible only for large $n$.
- A mix of informal and formal claims: Throughout the paper, even in the proofs, and especially in section 4, the paper mixes informal and formal results making it quite difficult to track what the result is stating. Approximate equalities $\approx$ and $=$ are used interchangeably and results that hold in the limit are taken to be valid for a finite T. This is especially felt in theorem 4.1 and lemma 4.1 and its corollaries. The writing and structure of the paper could really be improved.
- Tenuous links with practice and problems with solely analyzing generalization: The paper makes an effort to link its bounds with practice but is often unconvincing. For instance, concepts such as the "edge of stability" are discussed along with experiments simply because a step size and largest singular value appeared together in lemma 4.1 after two obscure assumptions such as commutation of a hessian and a covariance matrix and an ambiguous $\approx$.  For the trajectory based bounds, it is important to note that the obtained bounds are time dependent, meaning that a single pass over the data can quickly yield a vacuous result so this should be stated well before these bounds are used to try to justify practice. Putting aside the previous points, it is clear from Corollary 4.2 and its interpretation that solely analyzing generalization bounds can quickly loose meaning: yes an algorithm that stays close to initialization generalizes better but so does an algorithm that simply ignores the training data entirely. Over analyzing generalization bounds without thinking of the utility of the algorithm can lead to strange interpretations.
---
[1] Neu, Gergely, et al. "Information-theoretic generalization bounds for stochastic gradient descent."

[2] Wang, Z., & Mao, Y. (2021). On the generalization of models trained with SGD: Information-theoretic bounds and implications.

[3] Negrea, Jeffrey, et al. "Information-theoretic generalization bounds for SGLD via data-dependent estimates." Advances in Neural Information Processing Systems 32 (2019).

**Questions:**

Section 4 is difficult to understand. I can only make sense of the results if we exactly have $T = \infty$.
- Could you please formally restate the result showing that the noisy GD scheme you analyze converges to a mixture of Gaussians centered at local minima ? It is only briefly discussed with a mention to a result by Mandt et al.
- In lemma 4.1 are you assuming that there is only a single minimum per sample $S$ ?
- The results Corollary 4.2 and Theorem 4.2 appear to be algorithm independent as long as the algorithm converges to a distribution over local minima. Is this correct?

---

> ### Author Response · Authors · 2023-11-18
> **To Reviewer E5SG**
>
> We thank you sincerely for your constructive comments. Our responses follow.
>
> >- Some inappropriate comparisons and missing discussions: ... It is strange to say a bound on a different object is tighter.
>
> **Response.** The comparison with [1,2] is based on the observation that the SDE approximation exhibits nearly the same generalization error as SGD. Specifically, [1,2] invoke an auxiliary weight process, denoted as $\mathcal{A}\_{AWP}$, to obtain the generalization bound. In our paper, we use the SDE in Eq.(5), denoted as $\mathcal{A}\_{SDE}$. Let $\mathcal{A}\_{SGD}$ be the original algorithm of SGD. Recall that $\mathcal{E}$ is the generalization error, [1,2] provide the bound as follows:
> $$
> \mathcal{E}(\mathcal{A}\_{SGD})=\mathcal{E}(\mathcal{A}\_{SGD})+\mathcal{E}(\mathcal{A}\_{AWP})-\mathcal{E}(\mathcal{A}\_{AWP})\leq \text{GenBound}(\mathcal{A}\_{AWP})+|\mathcal{E}(\mathcal{A}\_{SGD})-\mathcal{E}(\mathcal{A}\_{AWP})|.
> $$
>
> Our paper presents the bound as:
> $$
> \mathcal{E}(\mathcal{A}\_{SGD})=\mathcal{E}(\mathcal{A}\_{SGD})+\mathcal{E}(\mathcal{A}\_{SDE})-\mathcal{E}(\mathcal{A}\_{SDE})\leq \text{GenBound}(\mathcal{A}\_{SDE})+|\mathcal{E}(\mathcal{A}\_{SGD})-\mathcal{E}(\mathcal{A}\_{SDE})|.
> $$
>
> In the experiments of this paper, given that $|\mathcal{E}(\mathcal{A}\_{SGD})-\mathcal{E}(\mathcal{A}\_{SDE})|$ is nearly negligible (particularly when compared with the scale of the GenBound) as shown in Figure 1 (and also in [4,5]), we thus directly compare $\text{GenBound}(\mathcal{A}\_{SDE})$ with the bound in [1,2].
>
> We have included these discussions in Appendix B.1 in the revision.
>
> [4] Wu et al. "On the noisy gradient descent that generalizes as sgd." ICML 2020.
>
> [5] Li et al. "On the validity of modeling sgd with stochastic differential equations (sdes)." NeurIPS 2021.
>
>
> >- More important are the missing discussions of other data dependent bounds ....
>
> **Response.** Note that our bounds in Thm 3.1-3.2 rely on Lemma 2.1, which, as clarified in [3], uses a distribution-dependent prior rather than a data-dependent one. Consequently, our developments align more closely with [6,1] rather than [3], and we do compare with [6,1] in Corollary 3.1 for the case of GLD.
>
> The previous SGLD bounds in [3] and several other prior works apply specifically to noisy SGD with a diagonal Gaussian covariance matrix. In this scenario, the noises are both data-independent and state-independent (i.e. independent of the current weights), so they can not directly be used to compare with our bounds. The appearance of "population gradient noise covariance" in [3] is a result of their construction of gradient incoherence (refer to Lemma D.3 in our revision). In summary, it arises in their bound due to mean differences between the Gaussian posterior and the data-dependent Gaussian prior, while in our bound, it emerges from both mean and covariance differences.
>
> In response to your suggestions, we also made an effort to incorporate the analysis from [3] into our work. Given that the noise term is now data-dependent and state-dependent, certain critical steps in [3] cannot be directly applied. Consequently, we use a variant of the bound from [3], and even in this case, additional assumptions are necessary to obtain the analytic form, as explained in Section D.7 for further details.
>
> [6] Pensia et al. "Generalization error bounds for noisy, iterative algorithms." ISIT 2018.
>
> >- Interpretability of the bounds: The choice of priors ... It is difficult to understand why this quantity is interesting as it is not well discussed.
>
> **Response.** Consider $n\gg b$. Recall the alignment term $\frac{\Sigma^\mu_tC_t^{-1}}{b}={\Sigma^\mu_t}{\Sigma_t}^{-1}$ where $\Sigma^\mu_t$ is the population gradient noise covariance (GNC) matrix, and $\Sigma_t$ is the GNC matrix for a specific training sample $S$. If these two matrices exhibit perfect alignment with each other *on average*, it implies that SGD (or its SDE approximation) is effectively performed in the true distribution of data, i.e., $w_t=w_{t-1}-\eta\nabla\mathbb{E}\_{Z}{\ell(w,Z)}+\eta\sqrt{\Sigma^\mu_t}N_t$. In other words, the perfect alignment of these two matrices indicates that SGD is insensitive to the randomness of $S$. Recall the key quantity in information-theoretic bounds, $I(W;S)$, which also measures the dependence of $W$ with the randomness of $S$, the term $\Sigma^{\mu}_{t}\Sigma^{-1}_t$ conveys a similar intuition in this context.

---

> > ### Author Response · Authors · 2023-11-18
> > **To Reviewer E5SG (cont.)**
> >
> > >- Moreover, there should be more discussion on ....
> >
> > **Response.** Thank you for this valuable suggestion; we have incorporated this into our limitations and future work discussion. We note that although deep neural networks are often overparameterized, recent research, as highlighted in works such as [7,8,9], reveals that SGD/GD only operates within a subspace of the neural networks, referred to as the *intrinsic dimension*. In this context, the SDE in Eq.(5) or $C_t$ itself can be defined within these invertible subspaces, and our derivations remain valid. Moreover, the theoretical characterization of the *intrinsic dimension* remains an open problem. The further development of this research direction will undoubtedly enhance and contribute to the advancements in our work.
> >
> > [7] Li et al. "Measuring the Intrinsic Dimension of Objective Landscapes." ICLR 2018.
> >
> > [8] Gur-Ari et al. "Gradient descent happens in a tiny subspace." arXiv preprint arXiv:1812.04754 (2018).
> >
> > [9] Larsen et al. "How many degrees of freedom do we need to train deep networks: a loss landscape perspective." ICLR 2022.
> >
> > >- A mix of informal and formal claims: ....
> >
> > **Response.** Thank you for pointing out this. We have revisited these parts and clarified that the majority of the results presented in Section 4 are asymptotic in nature. Please inform us if you have any further concerns or if additional improvements are needed in the wording.
> >
> > >- Tenuous links with practice and problems ...
> >
> > **Response.** Regarding the "edge of stability", here we give an intuitive explanation to the failure mode of $I(W;S)$ in the deterministic case (i.e. $I(W;S)\to\infty$) based on the interplay between the step size and the largest singular value. This is a by-product of our exploration and could be an avenue for further investigation. It's important to note that we do not aim to make overstated claims, and we have rephrased the relevant sections to prevent any potential confusion.
> >
> > >-  For the trajectory based bounds, it is important to note ...
> >
> > **Response.** We have provided more discussions about the time-dependent nature of trajetory-based bounds in the revision (refer to the highlighted text before Section 4). Moreover, we leverage this nature as motivation to introduce terminal-state-based bounds in Section 4. Essentially, if the preference is for time-independent results, we provide asymptotic results that are free from time dependence. In addition, we acknowledge the significance of adopting time-independent trajectory-based bounds, which, similar to Section 4, may require some additional assumptions, for example, smoothness, dissipativity, and regularity of the initial distribution used in a very recent study by [10]. It's important to note that empirically estimating certain parameters in time-independent bounds of [10] poses challenges.
> >
> > [10] Futami et al. "Time-Independent Information-Theoretic Generalization Bounds for SGLD." NeurIPS 2023.
> >
> > >- Putting aside the previous points ... .
> >
> > **Response.**  We agree that discussing generalization is meaningful only under the precondition that the algorithm has the satisfactory performance on training data. Our discussions following Corollary 4.2 have been revised accordingly.
> >
> > >- Could you please formally restate ... Gaussians centered at local minima ...
> >
> > **Response.** We have included this background information in Appendix B.3 of the revision. In the continuous case, this is from a classical result of the Ornstein–Uhlenbeck process, which is a Gaussian process. The stationary solution to its corresponding Fokker–Planck equation has a Gaussian density. While in the discrete case, in the long-time limit, $W_T$ represents the weighted sum of Gaussian noises. Note that in both cases, we need the quadratic approximation of loss (i.e. Eq.(7)).
> >
> > >- In lemma 4.1 are you assuming that there is only a single minimum ...?
> >
> > **Response.** We do not assume this. Lemma 4.1 is developed for a given $s$ and a given local minimal $w^*$ but $s$ itself could have more than one $w^*$. That is why we use  the phrase " when $w$ is close to any local minimal $w^*$" in the statement.
> >
> > >- The results Corollary 4.2 and Theorem 4.2 appear to be algorithm independent as long as the algorithm converges to a distribution ...
> >
> > **Response.** Corollary 4.2 and Theorem 4.2 are still algorithm-dependent. Note that that the expectations in these bounds are computed over the randomness of the local minimum $W_S^*$ for the training sample $S$, and the distribution $P_{W_S^*}$ is algorithm-dependent. For instance, it is well-known that numerous local minima exist, but SGD tends to exhibit implicit biases in selecting specific minima,  (e.g., flat minimal/low norm solution/ low rank solution), namely $P_{W_S^*}$ is non-uniform for SGD. As the exploration of the implicit biases of SGD is an ongoing research direction, gaining a better understanding of these biases may aid in more accurately characterizing $P_{W_S^*}$ in the future.

---

> > > ### Comment · Reviewer_E5SG · 2023-11-22
> > > **Continued response**
> > >
> > > - It is good that the limitations of the p.s.d assumption were included in the text.
> > >
> > > - On the terminal state results: In the paper, it sounds at first that you are stating that the stationary distribution of noisy GD (eq 5 ) on *any (or a large class)* loss is a mixture of gaussians on centered on the local minima. It is clear now that what you do is *first assume that the loss is a quadratic* then analyze the terminal state. I would put the quadratic approximation (eq 7) at the beginning of the section, before saying that stationary distributions are gaussians. The discussion of O.U in B.3 is unnecessary once this is clear. Unfortunately, this makes this section a bit less interesting as it's merely analyzing a noisy iterative scheme on a strongly convex quadratic loss.
> > >
> > > I chose to increase my score as modifications have been introduced and work has been put in making explicit the limitations. However, I still believe that the terminal state results are not properly presented and can erroneously lead the reader into believing stronger results are shown.

---

> > ### Comment · Reviewer_E5SG · 2023-11-22
> > **Response to rebuttal**
> >
> > Thank you for providing clarifications and for the modifications you have made on your paper.
> >
> > - I understood your strategy for analyzing SGD the way you describe it in your response.  Thank you for including this in the paper. I still believe for comparisons to make proper sense, the error terms (gap between SGD and SDE vs gap between AWP and SGD) need to be included explicitly and compared even if they are believed to be negligible. I understand that those error terms can only be quantified for specific settings, like bounded or lipschitz losses. So why not compare the bounds in that setting before stating one is tighter than the other.
> >
> > - Thank you for your clarifications on other data dependent bounds and covariance alignment.

---

> ### Author Response · Authors · 2023-11-22
> **Thanks for your reply!**
>
> We really appreciate the valuable discussions and constructive suggestions provided by the reviewer. These inputs have greatly contributed to improving our paper. Here, we response to two points raised by the reviewer.
>
> >- ... I understand that those error terms can only be quantified for specific settings, like bounded or lipschitz losses. So why not compare the bounds in that setting before stating one is tighter than the other.
>
> We attempted to make such a comparison in the case where the loss is $\beta$-smooth. While it is easy to obtain an upper bound for the residual term of $\mathcal{A}\_{\rm AWP}$, namely $\beta d\sum_{t=1}^T\sigma_t^2$ when using an isotropic Gaussian with variance $\sigma_t^2$ in the auxiliary weight process (and $d$ can be replaced by the trace of the sum of the noise covariance matrices in the anisotropic case), we still have some challenges in the analysis of the SDE process. This difficulty stems from the need for a similar analysis as the one used for the previous weak approximation results. Notably, the auxiliary weight process involves the same gradient signal as SGD for updating weights, simplifying the analysis of its cumulative noises. We acknowledge that, by the end of the Reviewer-Author discussion period, we may not be able to provide a concrete result. Nevertheless, we sincerely thank you for this suggestion, and we will keep it in mind.
>
> >- However, I still believe that the terminal state results are not properly presented and can erroneously lead the reader into believing stronger results are shown.
>
> We understand the reviewer's concern regarding the importance of avoiding any potential misinterpretation or overselling of results. To further avoid any potential misleading, we have made two additional modifications in the revised paper:
>
> 1) We have added additional discussions in the contributions part of the introduction  (cf. the last red texts on Page 2), we also state here for convience:
>
> *Comparing to the first family of bounds (i.e., trajectories-based bounds), the second family of bounds directly bound the generalization error via the terminal
> state, which avoids summing over training steps; these bounds can be tighter when the steady-state estimates are accurate. On the other hand, not relying on the steady-state estimates and the approximating
> assumptions they base upon is arguably an advantage of the first family.*
>
> 2) We have also clarified at the beginning of Section 4 that terminal state results depend on critical assumptions/approximations (including moving the quadratic loss to the first paragraph in Section 4):
>
> *To overcome the explicit time-dependence present in the bounds discussed in Section 3, one must introduce additional assumptions, with these assumptions being the inherent cost. For example, an important approximation used  in this section is the quadratic approximation of the loss...*
>
> We remain committed to refining the presentation of Section 4 and appreciate the opportunity to improve the clarity. Thank you again for your continued guidance.

---

### Author Response · Authors · 2023-11-18
**To All Reviewers**

We would like to thank all reviewers for the valuable feedback and constructive criticisms.  We have revised the paper accordingly (revisions presented in red fonts, for the ease of the reviewer) and will address your individual comments separately. Below is a summary of the revisions:

* We include more background in Appendix B covering the comparison with previous information-theoretic bounds for SGD, previous theoretical analysis validating SDE and the background on Gaussian distribution around local minimum.

* We add some discussions on the time-dependence of trajectories-based bounds before Section 4.

* We revise some theorem statements and discussions in Section 4.

* We add a new section on limitations and future work, where we discuss the dimension-dependence of the bounds.

* We add a data-dependent prior bound along with discussions in Appendix D.7.

* We add a summary table (i.e. Table 1) at the beginning of the Appendix to outline all the bounds presented in the main paper.

* We move the introduction of the notation system to Appendix A, and we remove our conclusion section due to space constraints.

We hope the revisions don't impose an undue workload to reviewers, and we appreciate your time and effort.

---

> ### Author Response · Authors · 2023-11-23
> **Latest Update Summary**
>
> Dear Reviewers,
>
> While we anticipated more interaction during the Reviewer-Author discussion period, we understand that the workload might be potentially heavy for the reviewers this year, and some may need to focus on their own rebuttals. Given the approaching end of the discussion, and the possibility of not being able to update the paper again, we provide a brief overview of two minor modifications in the latest revised version (in addition to the previous changes), resulting from further discussions with Reviewer E5SG:
>
> 1. We have incorporated additional discussions into the contributions parts of the introduction (refer to the last red-texted portion on Page 2). This addition aims to highlight the strengths and weaknesses of trajectories-based bounds and terminal-state-based bounds.
>
> 2. We have clarified at the beginning of Section 4 that terminal state results depend on some important assumptions/approximations, serving as a trade-off to eliminate time-dependence.
>
> Best,

---

### Meta-Review · Area_Chair_G2DW · 2023-12-14

**Metareview:**

Authors study (semi)-stochastic gradient descent where the noise is state dependent Gaussian. Authors use information theoretical tools to derive bounds on the expected generalization error. In addition, they also provide experiments to validate this algorithm is close to SGD.

This paper was reviewed by 4 reviewers and received the following Rating/Confidence scores: 6/2, 6/3, 6/4, 5/4. None of the reviewers championed the paper and they do have some valid concerns. Some of these include:
1- The overall organization can be done better.
2- Better motivating the paper.
3- Highlighting novelty and limitations.
4- Unclear technical novelty in the paper.

AC thinks that the paper has potential but requires significant revision, recommending reject for this ICLR. The decision is based upon the above weaknesses: None of the above weaknesses are severe when considered separately but together they decrease the quality of the paper significantly.

**Justification For Why Not Higher Score:**

I listed some improvement points in my MR.

**Justification For Why Not Lower Score:**

n/a

---

### Decision · Program_Chairs · 2024-01-16

Reject